# Meteor: Mamba-based Traversal of Rationale for Large Language and Vision Models

**Byung-Kwan Lee**
KAIST
leebk@kaist.ac.kr

**Chae Won Kim**
KAIST
chaewonkim@kaist.ac.kr

**Beomchan Park**
KAIST
bpark0810@kaist.ac.kr

**Yong Man Ro**
KAIST
ymro@kaist.ac.kr

## Abstract

The rapid development of large language and vision models (LLVMs) has been driven by advances in visual instruction tuning. Recently, open-source LLVMs have curated high-quality visual instruction tuning datasets and utilized additional vision encoders or multiple computer vision models in order to narrow the performance gap with powerful closed-source LLVMs. These advancements are attributed to multifaceted information required for diverse capabilities, including fundamental image understanding, real-world knowledge about common-sense and non-object concepts (*e.g.,* charts, diagrams, symbols, signs, and math problems), and step-by-step procedures for solving complex questions. Drawing from the multifaceted information, we present a new efficient LLVM, **M**amba-bas**e**d **t**rav**e**rsal **o**f **r**ationales (Meteor), which leverages multifaceted rationale to enhance understanding and answering capabilities. To embed lengthy rationales containing abundant information, we employ the Mamba architecture, capable of processing sequential data with linear time complexity. We introduce a new concept of *traversal of rationale* that facilitates efficient embedding of rationale. Subsequently, the backbone multimodal language model (MLM) is trained to generate answers with the aid of rationale. Through these steps, Meteor achieves significant improvements in vision language performances across multiple evaluation benchmarks requiring diverse capabilities, without scaling up the model size or employing additional vision encoders and computer vision models. Code is available in https://github.com/ByungKwanLee/Meteor.

## 1 Introduction

Following the successful zero-shot achievements of instruction-tuned large language models (LLMs) [1, 2], visual instruction tuning [3] has spurred the rapid development of large language and vision models (LLVMs). The emergence of closed-source LLVMs, such as GPT-4V [4], Gemini-Pro [5], and Qwen-VL-Plus [6], has prompted several studies to create high-quality question-answer visual instruction tuning datasets [6–11] and to scale up the model sizes of open-source LLVMs [10, 12–15], aiming to compete with their closed-source counterparts by leveraging the scaling law [16, 17, 2].

Recent research trends focus on enhancing image resolution [18, 6, 11, 19, 20] and dividing images into smaller sections [12, 14, 10, 21] to improve image perception capabilities. Additionally, some studies have utilized additional vision encoders [22–25] such as EVA-CLIP [26], DINOv2 [27], SAM [28], and SigLIP [29]. Various computer vision models [30–34] have also been employed

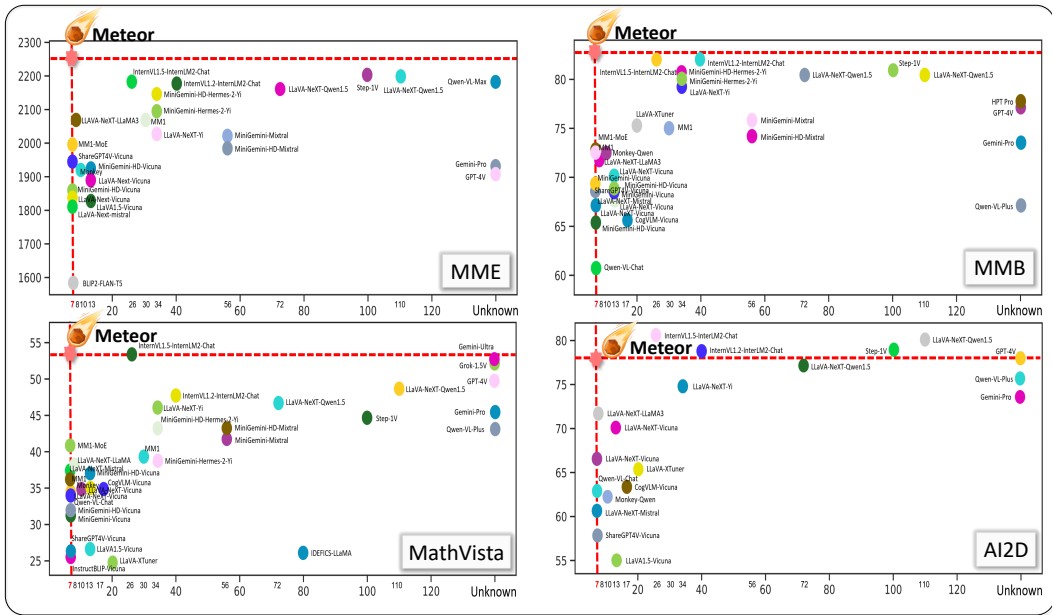

Figure 1: Across 7B to over 110B parameters, comparing lots of open- and closed-source LLVMs with Meteor on MME [35], MMB [36], MathVista [37], and AI2D [38] requiring diverse capabilities for image understanding, common-sense knowledge, non-object concept understanding, etc.

for tasks such as segmentation, detection, scene graph generation, and optical character recognition (OCR) to enhance LLVMs' answering capabilities with the help of external perception information.

These efforts, along with the curation of high-quality visual instruction datasets, have significantly reduced the performance gap between open- and closed-source LLVMs across numerous evaluation benchmarks and have even led to superior performances on some benchmarks. These successful developments are credited to the multifaceted information necessary for a wide range of capabilities. This encompasses fundamental image understanding, real-world knowledge of common-sense and non-object concepts (*e.g.,* charts, diagrams, symbols, signs, and math problems), and step-by-step procedures for solving complex questions.

Inspired by the key importance of the multifaceted information, we explore the possibility of designing efficient LLVMs that implicitly embed it as a form of multifaceted rationale (See Appendix A for more details), without significantly increasing model size and without using additional explicit vision encoders and computer vision models during the inference phase. Hence, we present a new efficient LLVM, **M**amba-bas**e**d **t**rav**e**rsal **of** **r**ationale (Meteor), comprising two core components: the Mamba architecture [39] and a multimodal language model (MLM) based on a pretrained large language model (LLM). The multifaceted rationale has rich information for achieving diverse capabilities, so its length is inherently long. This is why we employ the Mamba architecture, hereinafter referred to as Meteor-Mamba, which takes advantage of embedding lengthy input. It serves as an embedding module for the rationale, enabling Meteor-MLM, the MLM component, to address questions with the help of these embedded rationales. When conveying the knowledge of embedded rationales from Meteor-Mamba to Meteor-MLM, we introduce a new concept of *traversal of rationale*, which spurs embedding of long sequential rationales.

To ensure that Meteor encompasses diverse capabilities for vision-language tasks (e.g., image understanding, common-sense knowledge, charts, diagrams, documents, signs, symbols, and math problems), we gather 2.1M question-answer pairs from existing visual instruction tuning datasets: ShareGPT4V-Caption/Instruct [7], MiniGemini-Instruct [10], Doc-Downstream/Reason [20], GLLaVA-Align/Instruct [40], and Math-Vision/Instruct/Plus [41–43]. Subsequently, we utilize the light and fast Claude Haiku API [44] to generate detailed and comprehensive rationales tailored for the collected 2.1M question-answer pairs. These rationales are carefully filtered by human reviewers with the aid of GPT-4V, resulting in 1.1M question-rationale-answer triples (Appendix A).

Using the question-rationale pairs in the curated 1.1M triples, the first training step involves training Meteor-Mamba and miscellaneous projectors, *i.e.,* a vision projector and tor projector. During

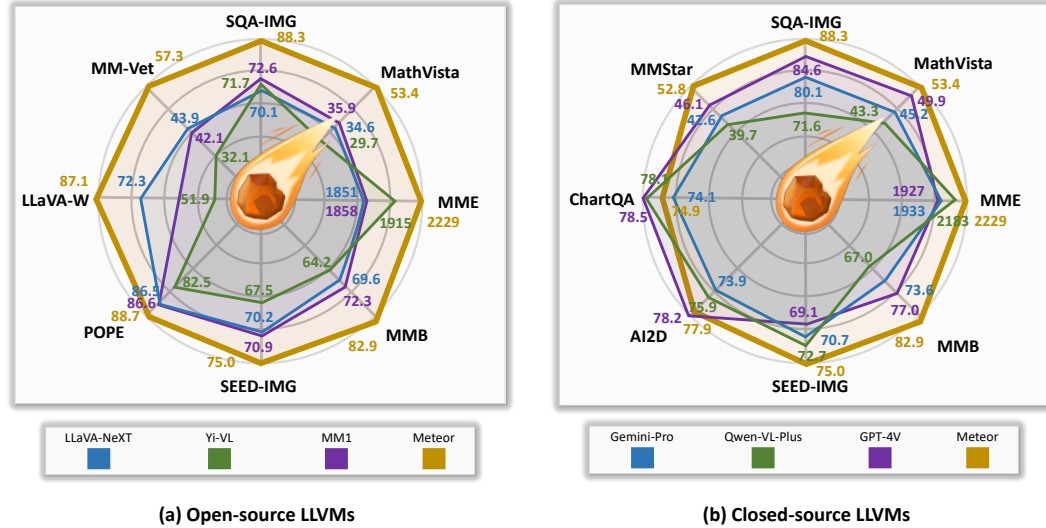

(a) Open-source LLVMs          (b) Closed-source LLVMs

Figure 2: Overall comparison of ☄ Meteor compared with other open- and closed-source LLVMs.

this step, Meteor-Mamba is trained to embed long sequential rationales. In the second training step, all components of ☄ Meteor are trained using the question-answer pairs in the curated 1.1M triples. Through these steps, we demonstrate that ☄ Meteor significantly improves vision-language performance, as shown in Figure 1, compared with other open- and closed-source LLVMs, on numerous benchmarks requiring diverse capabilities. As illustrated in Figure 2, these results advocate for the possibility of building efficient LLVMs with a multifaceted rationale, beyond the scope of scaling model size, additional vision encoders, and multiple computer vision models.

Our contribution can be summarized into two main aspects:

- We introduce a new efficient large language and vision model (LLVM), **M**amba-bas**e**d **t**rave**r**sal **o**f **r**ationale (☄ **Meteor**), which comprehends long sequential rationales under a new concept of *traversal of rationale* and predict answers with the help of the rationale.

- Despite its efficient model size, ☄ **Meteor** showcases significant advancements across various evaluation benchmarks requiring diverse capabilities for image understanding, common-sense, non-object concepts, and more.

## 2 Related Works

**Rationale-Guided Prediction.**   Behind the answers of large language models (LLMs), rationale has played a significant role in enhancing answering capabilities across a myriad of natural language processing and vision language tasks in various forms such as (a) human annotation [45, 46], (b) knowledge distillation [47–51], and (c) chain-of-thought (CoT) [52–55]. Rationale mimics the human thought process, providing explanations or justifications before answering questions. (a) Strout *et al.*[45] and Lu *et al.*[46] integrate human-annotated rationale with LLMs to fortify their performance, thereby enhancing model robustness against out-of-distribution scenarios with human annotations. (b) In knowledge distillation, rationale is used to effectively distill LLMs [47–51] into smaller language models. They first extract rationale from LLMs and then fine-tune smaller language models with the extracted rationale, demonstrating their efficacy through various evaluation benchmarks. (c) To directly apply rationale to LLMs, researchers have used a 'think step-by-step' prompt called Chain of Thought (CoT) [52]. When combined with few-shot learning, CoT elicits step-by-step rationale directly from LLMs for an input question followed by a series of questions, rationales, and answers used as few-shot examples. This has been further streamlined by automating rationale generation in CoT prompts, as explored by [53] and [54], eliminating the need for human-annotated rationale for few-shot examples.

Similar to (b), we also leverage the power of closed-source LLVMs but, in contrast, employ another model, Meteor-Mamba, to embed the rationale, instead of directly training Meteor-MLM to generate the created rationales. In other words, we separate the roles of the models where Meteor-MLM generates the answer, while Meteor-Mamba embeds the rationale, which encompasses diverse

capabilities: fundamental image understanding, incorporation of real-world knowledge of common-sense, understanding of non-object concepts (e.g., charts, diagrams, symbols, signs, and math), and following systematic step-by-step procedures for solving complex questions. Once accompanied by LLVMs, their answer capabilities are expected to improve with the help of the rationale.

**Large Language and Vision Models.** Following the emergence of visual instruction tuning datasets created by LLaVA [3, 8, 12] and InstructBLIP [9], there has been rapid development of large language and vision models (LLVMs): Shikra [56], IDEFICS [57], Qwen-VL [6], MiniGPT-4 [58], Otter [59], mPLUG-Owl [60, 19], ShareGPT4V [7], LLaVA-XTuner [61], Intern-XC [62], MM1 [14], MiniGemini [10], InternVL Families [15, 63], along with efforts to gather or curate high-quality visual instruction tuning datasets for various purposes: ShareGPT4V [7], ALLaVA [64], MiniGemini [10], mPLUG-DocOwl [20], GLLaVA [40], MathVision [41], MathInstruct [42], and MathPlus [43]. Recently, Otter-HD [18], Qwen-VL [6], CogVLM [11], and mPLUG Families [19, 20] have increased image resolution. Additionally, LLaVA-NeXT [12], MM1 [14], and MiniGemini [10] divide images into smaller sections, while LLaVA-UHD [21] and InternVL1.5 [63] employ dynamically split image partitions depending on their sizes. These research trends aim to improve image perception capabilities, thereby enhancing understanding of images and natural language instructions. Furthermore, BRAVE [22], DeepSeek-VL [23], OmniFusion [24], MoVA [65], and AM-RADIO [25] have utilized additional vision encoders such as EVA-CLIP [26], DINOv2 [27], SAM [28], and SigLIP [29]. Apart from those, SpatialVLM [30], ASMv2 [31], LAR/LAF [32], CoLLaVO [33], and MoAI [34] employ multiple computer vision models for tasks such as segmentation, detection, scene graph generation, and optical character recognition (OCR).

We view this series of efforts as procedures for enlarging LLVM's knowledge space regarding multifaceted information. From a fresh perspective, we believe that embedding it in the form of a multifaceted rationale serves as a key in developing efficient LLVMs, where they include in-depth explanations required for acquiring diverse capabilities. From this perspective, Meteor is expected to inherently embed the multifaceted rationale and improve answering capabilities with the help of the embedded rationale, even without significantly increasing model size or relying on additional explicit vision encoders and computer vision models.

# 3  Meteor: Mamba-based traversal of rationale

**Model Architecture.** As illustrated in Figure 3, Meteor comprises a vision encoder, vision projector, Mamba architecture [39], tor projector, and backbone multimodal language model (MLM) based on a pretrained large language model (LLM). For the vision encoder, we use CLIP-L/14 [66], which is a text-aligned vision module that takes advantage of image understanding adeptness powered by text descriptions. For the vision projector and tor projector, we employ MLP modules containing two fully-connected layers with the GELU activation function [67]. Next, we use the Mamba-130M architecture for computational efficiency and adopt InternLM2-7B [68, 69], learned with 2T tokens of multilingual text data in RLHF [70], as the backbone large language model (LLM).

**Configuration of Visual Instruction Tuning Dataset.** To build a visual instruction tuning set, we cover not only fundamental image understanding but also a wide range of diverse capabilities: common-sense knowledge, non-object concepts (*e.g.,* charts, diagrams, documents, signs, symbols, and math problems), cognitive reasoning, multi-discipline tasks, and integrated abilities. For the question-answer visual instruction tuning dataset, we choose 664K question-answer pairs in ShareGPT4V-Instruct [7], including LLaVA-Instruct-665K [8]. Additionally, in ShareGPT4V-Caption [7], we select 91K image descriptions for images from LAION [71], CC [72], SBU [73], MS-COCO [74], TextCaps [75], and web images [76–78] that depict landmarks, animals, celebrities, art, text, and nature. The selected question-answer pairs primarily focus on fundamental image understanding and common-sense knowledge, with fewer data samples covering non-object concepts, cognitive reasoning, multi-discipline tasks, and integrated abilities. To strengthen these areas, we selectively gather 27K question-answer pairs of DocVQA [79], ChartQA [80], DVQA [81], and AI2D [38] from MiniGemini-Instruct [10]. Moreover, we use 574K/27K question-answer pairs of DeepForm [82], InfoVQA [83], DocVQA [79], KleisterCharity [84], TabFact [85], TextVQA [86], WikiTable [87], TextCaps [75], and VisualMRC [88] from Doc-Downstream/Reason [20]. To achieve

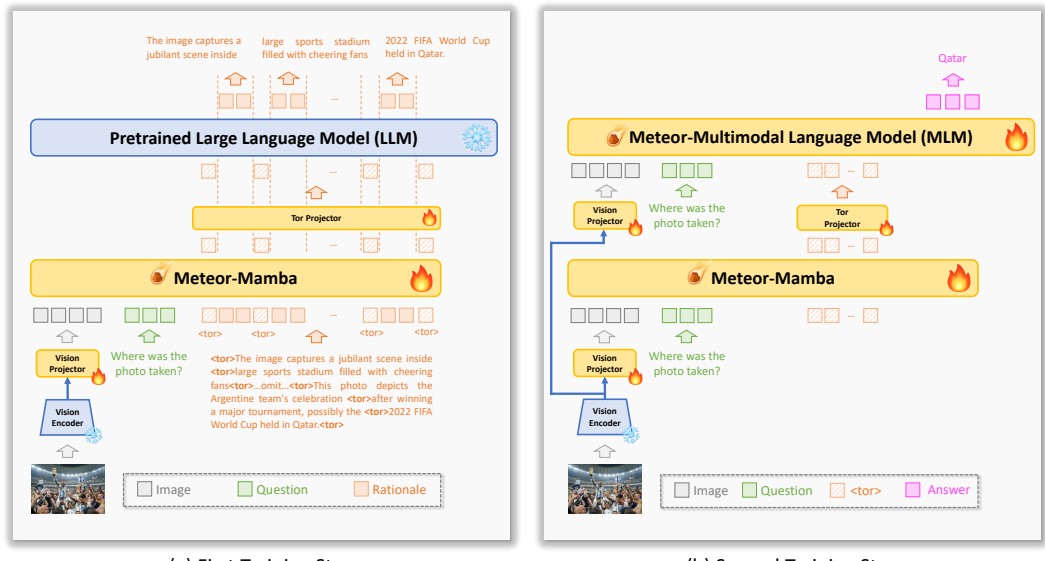

(a) First Training Step  (b) Second Training Step

Figure 3: Overview of 🌠 Meteor architecture and its training steps. Note that, 'Meteor-Multimodal Language Model (MLM)' indicates that as training progresses, the pretrained language model evolves into a multimodal language model.

broad coverage of math knowledge, we also include 177K GLLaVA-Align/Instruct [40], 3K Math-Vision [41], and 566K text-only samples from Math-Instruct/Plus [42, 43]. In summary, we collect 755K real-world images, 627K images for documents, charts, diagrams, signs, and symbols, and 747K math samples (180.5K with images and 566.8K text-only). Overall, the question-answer visual instruction tuning samples sum up to 2.1M.

**Curating Rationale.**    Using the gathered 2.1M question-answer pairs, we utilize the light and fast Claude Haiku API [44] to generate detailed and comprehensive rationales. We use the prompt template: "*Question:* {}. *Answer:* {}. *Based on the question and answer, carefully provide an explanation about how to answer the question in detail.*" Here, {} represents the placeholder for the corresponding language description. Afterward, we assess the rationale score from GPT-4V [4] using the following template: "*Question:* {}. *Rationale:* {}. *Answer:* {}. *Based on the question, rationale, and answer, provide a score from 0 to 10, evaluating how well the rationale is described to solve the question. If the given rationale is insufficient, you should rigorously give a score below 5.*" Subsequently, we filter out the generated rationales with a score below 5. The rationales that pass this automated evaluation are then subjected to human review to determine *Yes or No* on whether they provide a proper description to address the question. Finally, this series of processes yields 1.1M question-rationale-answer triples, which include 338K real-world images covering common-sense knowledge and a few samples for diverse capabilities, 379K images for documents, charts, diagrams, signs, and symbols, and 342K math samples (165K with images and 177K text-only).

**Mamba Architecture.**    To make LLVMs inherently possess rationale when addressing complex questions, we generate comprehensive rationales based on question-answer pairs. Subsequently, we employ the Mamba architecture [39], leveraging its capability to handle lengthy rationales while maintaining computational efficiency. This approach allows us to effectively incorporate the rationale in an environment where the curated 1.1M question-rationale pairs have an average length of 213 tokens, which is approximately ten times longer than the average length of 22 tokens of ground truth answers in typical visual instruction tuning datasets [21, 7].

**Traversal of Rationale.**    However, it is crucial to note that we cannot acquire and utilize rationales during the inference phase without API-based models, since only user questions are given. Therefore, we propose a new concept called *traversal of rationale* to effectively provide the rationale to Meteor-MLM without any help from external APIs in the inference phase. Inspired by retrieval-based knowledge [89], we introduce a special token, <tor> (stands for **t**raversal **o**f **r**ationale), and evenly distribute 10 fixed number of <tor> tokens, as described in Figure 3. The rationale planted with <tor> is propagated into Meteor-Mamba along with image and question tokens, and then the output

| LLVMs | Q-Bench | SQA[I] | AI2D | ChartQA | SEED[I] | POPE | HallB | MME | MathVista | MMB | MMB[CN] | MM-Vet | LLaVA[W] |
|---|---|---|---|---|---|---|---|---|---|---|---|---|---|
| BLIP2-13B [90] | - | 61.0 | - | - | 46.4 | 85.3 | - | 1584 | - | - | - | 22.4 | - |
| InstructBLIP-7B [9] | 56.7 | 60.5 | - | - | 53.4 | - | 53.6 | - | 25.3 | 36.0 | 23.9 | 26.2 | - |
| InstructBLIP-13B [9] | - | 63.1 | - | - | - | 78.9 | - | - | - | 33.9 | - | 25.6 | - |
| IDEFICS-9B [57] | 51.5 | - | - | - | - | 74.6 | - | 1353 | 19.8 | 48.2 | 25.2 | 23.7 | - |
| Qwen-VL-7B [6] | 59.4 | 67.1 | - | - | - | - | - | - | - | 38.2 | 7.4 | - | - |
| Qwen-VL-Chat-7B [6] | 33.8 | 68.2 | - | - | 58.2 | - | 56.4 | 1849 | - | 60.6 | 56.7 | 47.3 | - |
| MiniGPT-4-7B [58] | 51.8 | - | - | - | - | - | - | - | 23.1 | 23.0 | 11.9 | 22.1 | - |
| Otter-7B [59] | 47.2 | - | - | - | - | 72.5 | - | 1599 | 19.7 | 48.3 | - | 24.7 | - |
| LLaVA-7B [3] | - | 38.5 | - | - | - | 80.2 | 44.1 | 1055 | - | 34.1 | 14.1 | 26.7 | - |
| LLaVA1.5-7B [8] | 60.1 | 66.8 | - | - | 58.6 | 85.9 | - | 1805 | - | 64.3 | 58.3 | 30.5 | 63.4 |
| LLaVA1.5-13B [8] | 61.4 | 71.6 | 54.8 | 18.2 | 61.6 | 85.9 | 46.7 | 1826 | 27.6 | 67.7 | 63.6 | 35.4 | - |
| mPLUG-Owl-7B [60] | 58.9 | - | - | - | - | - | - | - | 22.2 | 46.6 | - | - | - |
| mPLUG-Owl2-7B [19] | 62.9 | 68.7 | - | - | - | - | - | - | - | 64.5 | 60.3 | 36.2 | - |
| ShareGPT4V-7B [7] | 63.4 | 68.4 | - | - | 69.7 | - | 49.8 | 1944 | 25.8 | 68.8 | 62.2 | 37.6 | - |
| InternLM-XC-7B [62] | 64.4 | - | - | - | 66.1 | - | 57.0 | 1919 | 29.5 | 74.4 | 72.4 | 35.2 | - |
| Monkey-10B [91] | - | 69.4 | - | - | 68.9 | - | 58.4 | 1924 | 34.8 | 72.4 | 67.5 | 33.0 | - |
| VILA-7B [92] | - | 68.2 | - | - | 61.1 | 85.5 | - | - | - | 68.9 | 61.7 | 34.9 | - |
| VILA-13B [92] | - | 73.7 | - | - | 62.8 | 84.2 | - | - | - | 70.3 | 64.3 | 38.8 | - |
| SPHINX-7B [93] | - | 70.6 | - | - | 71.6 | 86.9 | - | 1797 | 27.8 | 65.9 | 57.9 | 40.2 | - |
| SPHINX-MoE-7B×8 [94] | 66.2 | 70.6 | - | - | 73.0 | 89.6 | - | 1852 | 42.7 | 71.3 | - | 40.9 | - |
| SPHINX-Plus-13B [94] | 66.2 | 70.6 | - | - | 74.8 | 89.1 | 52.1 | 1741 | 36.8 | 71.0 | - | 47.9 | - |
| LLaVA-NeXT-7B [12] | - | 70.1 | - | - | 70.2 | 86.5 | - | 1851 | 34.6 | 69.6 | 63.3 | 43.9 | 72.3 |
| LLaVA-NeXT-8B [12] | - | - | 71.6 | 69.5 | - | - | - | 1972 | 37.5 | 72.1 | - | - | 80.1 |
| LLaVA-NeXT-13B [12] | - | 73.6 | 70.0 | 62.2 | 72.2 | 86.7 | - | 1892 | 35.1 | 70.0 | 68.5 | 47.3 | 72.3 |
| MM1-7B [14] | - | 72.6 | - | - | 69.9 | 86.6 | - | 1858 | 35.9 | 72.3 | - | 42.1 | - |
| MM1-MoE-7B×32 [14] | - | 74.4 | - | - | 70.9 | 87.8 | - | 1992 | 40.9 | 72.7 | - | 45.2 | - |
| MiniGemini-HD-7B [10] | - | - | - | - | - | - | - | 1865 | 32.2 | 65.8 | - | 41.3 | - |
| MiniGemini-HD-13B [10] | - | - | - | - | - | - | - | 1917 | 37.0 | 68.6 | - | 50.5 | - |
| Meteor-7B | **69.2** | **88.3** | **77.9** | **74.9** | **75.0** | **88.7** | **60.4** | **2229** | **53.4** | **82.9** | **82.1** | **57.3** | **87.1** |

Table 1: Comparison with the current existing open-source LLVMs, evaluating vision language performances of 🌠Meteor on numerous evaluation benchmarks requiring diverse capabilities: Q-Bench [95], SQA[I] [96], AI2D [38], ChartQA [80], SEED[I] [97], POPE [98], HallB [99], MME [35], MathVista [37], MMB [36], MMB[CN] [36], MM-Vet [100], and LLaVA[W] [3]. Note that, AI2D and ChartQA performances for LLaVA family models are evaluated under zero-shot conditions, while 🌠Meteor uses training dataset for them.

features in Meteor-Mamba are directly propagated into Meteor-MLM. Here, we autoregressively train Meteor to generate the part of the rationale between <tor>, whenever Meteor sees the special token <tor>. This procedure ensures that each <tor> represents the following rationale part until the next <tor> is encountered. Using a single <tor> token to encompass the rationale may not work well when embedding lengthy rationales, and if we do not consider distributing <tor> tokens in the rationale, then the later token does not refer to the earlier ones well due to the common problem of the autoregressive mechanism's forgetting nature [39]. This is why we place multiple <tor> tokens in the rationale instead of one.

**Training Strategy.** In the first training step, we leverage the question-rationale pairs in the curated 1.1M triples to train Meteor-Mamba and miscellaneous projectors. Throughout this step, the long sequential rationale is embedded into Meteor-Mamba through traversal of rationale by autoregressively generating rationale parts between the special tokens <tor>. By freezing Meteor-MLM, Meteor-Mamba seamlessly incorporates the rationale. In the second training step, we utilize the question-answer pairs in the curated 1.1M triples to jointly train Meteor-Mamba, Meteor-MLM, and the miscellaneous projectors. Here, multiple <tor> special tokens are only propagated to Meteor-Mamba. Then, the rationale-embedded features in Meteor-Mamba corresponding to the special tokens <tor> are only fed into Meteor-MLM, enabling it to adeptly answer complex questions, even in the absence of explicit rationale descriptions. In essence, these steps equip 🌠Meteor with the capability to effectively address complex questions with the aid of the rationale.

# 4 Experiment

**Implementation Detail.** To ensure successful reproducibility, we outline three crucial technical details of 🌠Meteor: the structure of (a) Meteor-Mamba and Meteor-MLM, (b) vision encoder and miscellaneous projectors, and (c) training and inference details.

**(a)** To build Meteor-Mamba, we use the Mamba architecture with 24 layers and a 768 hidden dimension, resulting in a total of 130M parameters, which is relatively trivial compared to the approximately 7B parameters of the pretrained InternLM2-7B [68, 69]. It is executed under the

| LLVMs | CP | FP | IR | LR | ST | MA | Avg |
|---|---|---|---|---|---|---|---|
| Yi-VL-34B [13] | 53.2 | 31.2 | 52.0 | 32.4 | 12.4 | 35.2 | 36.1 |
| CogVLM-Chat-17B [11] | 66.8 | 36.8 | 49.2 | 31.2 | 23.6 | 11.6 | 36.5 |
| SPHINX-MoE-7B×8 [94] | 58.4 | 40.8 | 47.6 | 35.2 | 19.2 | 32.0 | 38.9 |
| InternVL1.2-40B [15] | 67.6 | 43.2 | 61.2 | 47.2 | 24.0 | 19.2 | 43.7 |
| LLaVA-NeXT-34B [12] | 66.4 | **52.0** | 62.4 | 46.0 | 32.4 | **53.6** | 52.1 |
| Meteor-7B | **69.6** | 45.6 | **63.6** | **53.2** | **42.0** | 42.8 | **52.8** |

(a) MMStar [101]

| LLVMs | TD | TL | TO | VI | VD | VO | Avg |
|---|---|---|---|---|---|---|---|
| G-LLaVA-7B [40] | 20.9 | 20.7 | 21.1 | 17.2 | 16.4 | 9.4 | 16.6 |
| LLaVA-NeXT-13B [12] | 12.8 | 12.0 | 9.9 | 10.7 | 9.7 | 6.3 | 10.3 |
| ShareGPT4V-13B [7] | 16.2 | 16.2 | 6.6 | 15.5 | 13.8 | 3.7 | 13.1 |
| SPHINX-Plus-13B [94] | 13.9 | 11.6 | 14.9 | 11.6 | 13.5 | 10.4 | 12.2 |
| SPHINX-MoE-7B×8 [94] | **26.2** | 17.4 | 26.7 | 16.7 | 12.5 | 11.1 | 16.8 |
| Meteor-7B | 25.5 | **21.7** | **27.4** | **21.7** | **19.2** | **14.7** | **21.7** |

(b) MathVerse [102]

| Benchmarks | OmniFusion [24] | DeepSeek-VL [23] | MoVA [22] | ASMv2 [31] | LAF [32] | CoLLaVO [33] | MoAI [34] | Meteor |
|---|---|---|---|---|---|---|---|---|
| POPE | 87.2 | 88.1 | 88.6 | 86.3 | **88.8** | 87.2 | 87.1 | 88.7 |
| SQA-IMG | 69.2 | 57.7 | 74.4 | 87.1 | - | 80.7 | 83.5 | **88.3** |
| LLaVA-W | - | - | - | 78.9 | - | 69.5 | 71.9 | **87.1** |
| MM-Vet | 39.4 | 41.5 | - | 41.3 | 38.9 | 40.3 | 43.7 | **57.3** |
| MMStar | - | - | - | - | - | 42.1 | 48.7 | **52.8** |

(c) Comparison with LLVMs using additional vision encoders and computer vision models

| LLVMs | Recognition | OCR | Knowledge | Language Generation | Spatial Awareness | Math Problems | Avg |
|---|---|---|---|---|---|---|---|
| CoLLaVO-7B [33] | 45.6 | 31.1 | 29.8 | 30.2 | 37.9 | 5.8 | 41.0 |
| MoAI-7B [34] | 48.3 | 34.8 | 33.5 | 33.0 | 39.7 | 7.7 | 43.7 |
| Meteor-7B w.o. Meteor-Mamba | 44.5 | 33.5 | 41.8 | 31.3 | 38.6 | 29.2 | 44.8 |
| Meteor-7B | **54.1** | **60.1** | **44.2** | **45.0** | **59.3** | **57.7** | **57.3** |

(d) Evaluating sub-benchmarks in MM-Vet [100] with LLVMs utilizing computer vision models

Table 2: Detailed comparison of 🔥Meteor across more challenging evaluation benchmarks.

efficient computation of hardware-aware state expansion [39], where we borrow the tokenizer [103] from the backbone MLM to fit the language expression space in the backbone MLM. Meteor-MLM is based on InternLM2-7B [68, 69] with 32 layers and a 4096 hidden dimension.

**(b)** We use a vision encoder with 428M CLIP-L/14 [66], which has 24 layers and a 1024 hidden dimension. The resolution of the positional embedding is interpolated from $24 \times 24$ to $35 \times 35$ to accommodate a $490 \times 490$ image resolution. The vision projector involves an MLP that adapts the hidden dimension from 1024 to 4096 to fit that of the backbone MLM. Similarly, we build the tor projector to convey embedded rationales from Meteor-Mamba into Meteor-MLM, employing the same structure as the vision projector but transferring the hidden dimension from 768 to 4096.

**(c)** We train and evaluate 🔥Meteor in the following computing environment: Intel(R) Xeon(R) Gold 6230, 256 GB RAM, and 8×NVIDIA RTX A6000 48GB VRAM. To efficiently train it, we use one epoch of training for each training step under 4-bit quantization and bfloat16 data type [104] for Meteor-MLM, where double quantization and normalized float 4-bit (nf4) [105] are used. Meteor-Mamba uses float32 data type because training it with bfloat16 or float16 has been reported to produce an unstable learning process. In addition, QLoRA [106, 105] is used to train Meteor-MLM, with 64 rank and 64 alpha parameters. We use the AdamW [107] optimizer and schedule the learning rate by cosine annealing [108] from 1e-4 to 1e-6 in each training step, with gradient checkpointing [109] applied to Meteor-MLM for efficient memory management. With a gradient accumulation of 6, we set batch sizes of 192 and 576 for each training step, and each step takes approximately three days. For efficient inference, 🔥Meteor is validated in 4-bit quantization, and we use deterministic beam search ($n = 3$)[110] for text generation. Note that we implement not only Meteor-MLM but also numerous baselines under the efficient propagation from FlashAttention2[111, 112].

**Evaluation.** We have evaluated 🔥Meteor on numerous vision-language benchmarks, the details of which are described in Appendix B. These benchmarks require multifaceted information for diverse capabilities, including fundamental image understanding, real-world knowledge of common-sense knowledge, charts, diagrams, documents, signs, symbols, math problems, and more. Figure 1-2 and Table 1 illustrates vision language performances of various LLVMs, including Meteor-7B, open-, and closed-source LLVMs with various sizes. It is noteworthy that Meteor-7B noticeably outperforms the other models, demonstrating its efficacy and efficiency in using embedded multifaceted rationales from Meteor-Mamba. The detailed generation quality of 🔥Meteor is described in Appendix C. Apart from the results in Table 1, those in Table 2 signify that Meteor-7B also excels at more challenging benchmarks, which require multifaceted information simultaneously. Meteor-7B has outperformed

| Arch | Param | BPS | MMB | MM-Vet |
|------|-------|-----|-----|--------|
| BERT-B | 110M | 71 | 80.6 | 53.6 |
| GPT2-S | 117M | 62 | 80.9 | 53.5 |
| XLNet-B | 110M | 56 | 81.6 | 53.9 |
| Mamba | 130M | **118** | **82.9** | **57.3** |

(a) Meteor-Mamba

| LLMs | Param | MMB | MM-Vet |
|------|-------|-----|--------|
| Vicuna1.5 | 7B | 80.1 | 53.8 |
| LLaMA2 | 7B | 78.8 | 51.6 |
| LLaMA3 | 8B | 81.7 | 56.0 |
| InternLM2 | 7B | **82.9** | **57.3** |

(b) Meteor-MLM

| Num | MMB | MM-Vet |
|-----|-----|--------|
| #2 | 76.1 | 47.9 |
| #5 | 82.2 | 55.8 |
| #10 | **82.9** | **57.3** |
| #15 | 82.8 | **57.3** |

(c) Number of <tor> tokens

| Position | MMB | MM-Vet |
|----------|-----|--------|
| Start | 79.8 | 48.2 |
| End | 74.2 | 45.1 |
| Random | 74.3 | 45.4 |
| Even | **82.9** | **57.3** |

(d) Position of <tor> tokens

| Mamba | Rationale | MMB | MM-Vet |
|-------|-----------|-----|--------|
| ✗ | ✗ | 73.2 | 44.8 |
| ✗ | ✓ | 77.0 | 48.2 |
| ✓ | ✗ | 74.0 | 45.9 |
| ✓ | ✓ | **82.9** | **57.3** |

(e) Mamba & Rationale

| $\mathcal{Q}$-$\mathcal{R}$ | MME | MMB | MM-Vet |
|------|-----|-----|--------|
| 0% | 1989 | 74.0 | 45.9 |
| 30% | 2105 | 77.5 | 51.7 |
| 60% | 2218 | 82.5 | 56.8 |
| 90% | **2229** | **82.9** | **57.3** |

(f) Ratio of $\mathcal{Q}$-$\mathcal{R}$ in training

Table 3: Ablation studies to identify the effectiveness of Meteor-Mamba and rationale through traversal of rationale by controlling the six main factors.

other existing models by a large margin, some of which are equipped with additional vision encoders or computer vision models, demonstrating that rationales can provide multifaceted information more effectively than enhanced visual perception.

**Ablation Studies.** Furthermore, we have conducted several ablation studies to securely corroborate the effectiveness of our proposed method in light of six factors: (a) Meteor-Mamba, (b) Meteor-MLM, (c) the number of <tor> special tokens, (d) the distribution of <tor> special tokens, (e) rationale, and (f) 1.1M question-rationale-answer triples. Table 3 shows the following findings. Note that, Appendix D represents further ablation studies.

**(a)** Once Meteor-Mamba is replaced with other transformer-based models: BERT [113], GPT2 [114], and XLNet [115], we have discovered that the Mamba architecture takes advantage of its efficiency for embedding multifaceted rationales in terms of both computational complexity and model size. As the results suggest, Mamba demonstrates the highest batches-per-second (BPS) value and zero-shot performances on MME and MMB benchmarks among architectures of similar sizes, enabled by its inherent computational efficiency based on linear complexity and strong long sequence modeling capability [39], which other Transformer-based model architectures lack.

**(b)** We have tried using various pretrained LLMs of comparable sizes for Meteor-MLM, in order to identify the effectiveness of embedding multifaceted rationale together with traversal of rationale. We have observed that InternLM2 [69] has shown the best performances.

**(c)** Varying the number of <tor> special tokens from 2 to 15, we have optimized it based on the vision language performances. The results suggest that using 10 <tor> special tokens shows the best performances for embedding abundant multifaceted rationales, balancing between compression and information preservation.

**(d)** The performances of ☄Meteor depend on the distribution of <tor> special tokens when Meteor-Mamba is trained to embed multifaceted rationales. Given the observations, evenly distributing the tokens across lengthy rationales has shown the best performances. Prepending them to lengthy rationales may hinder effective embedding due to forgetting nature, and appending them to rationales in the end may not be guaranteed to understand the rationale. Randomly distributing the tokens across rationales may disrupt Meteor-Mamba's ability to stably learn the pattern of embedding rationales. Conversely, evenly distributed <tor> special tokens can segment lengthy rationales into shorter chunks and progressively embed them in a consistent manner, avoiding the issues of other distributions.

**(e)** In order to prove the effectiveness of multifaceted rationales through Meteor-Mamba, we ablated the use of the Mamba architecture and the rationales. The first row represents baseline performances where backbone MLM is only trained. For the second row, we only train backbone MLM with the curated multifaceted rationales without Meteor-Mamba, and for the third row, Mamba has been trained to embed answer instead of the rationales. Compared to the last row where Meteor is evaluated, the second and third rows fall short of performances, clearly showing that using multifaceted rationales through the Mamba architecture has contributed to performance improvement.

**(f)** As another way of showing the significance of multifaceted rationales, we have trained ☄Meteor with different amounts of question-rationale pairs and evaluated ☄Meteor trained with each of them.

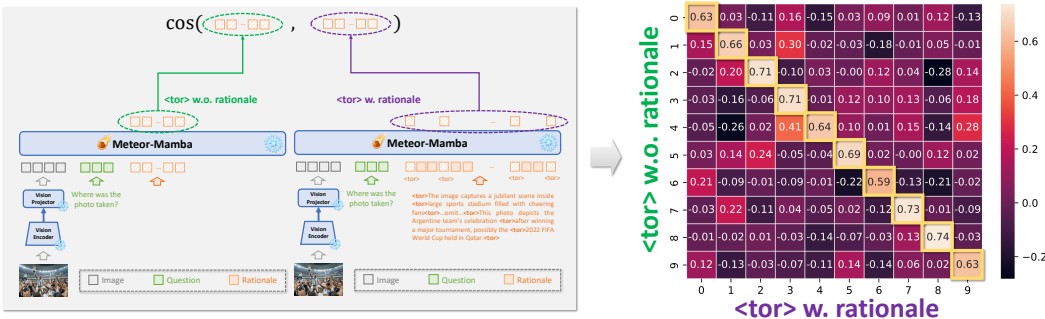

Figure 4: Illuminating how the feature correspondences of cosine similarity are computed under the trained Meteor-Mamba, and showing the feature disparity for <tor> with/without rationale.

As expected, the more question-rationale pairs used in first training step, the better performances achieves, demonstrating the significance of utilizing multifaceted rationales for diverse capabilities.

**Meteor-Mamba's Ability to Embed Rationales.** We conduct a thorough analysis to confirm that Meteor-Mamba effectively embeds the rationales. To do this, we perform a retrieval task for multifaceted rationales, where we prepare ten different question-rationale pairs $(\mathcal{Q}_i, \mathcal{R}_i)$ where $i = 0, 1, \cdots, 9$. These pairs are propagated through Meteor-Mamba with or without rationales under <tor> special tokens. This results in two sets of output features: one with rationale $z_i^{\text{w.}}$ and one without rationales $z_j^{\text{w.o.}}$, with $j = 0, 1, \cdots, 9$. We extract features corresponding to the placement of <tor> tokens, resulting in $\mathbf{z}_i^{\text{w.}} \in \mathbb{R}^{10 \times 768}$ and $\mathbf{z}_j^{\text{w.o.}} \in \mathbb{R}^{10 \times 768}$, where the dimension 10 corresponds to the number of <tor> tokens. We then compute the cosine similarity between $\mathbf{z}_i^{\text{w.}}$ and $\mathbf{z}_j^{\text{w.o.}}$ to measure the similarity of their representations. As illustrated in Figure 4, the diagonal values in the cosine similarity matrix are much higher than the off-diagonal values. This result indicates that Meteor-Mamba successfully embeds the rationale, and its output features contain multifaceted information even without explicit rationales in natural language. This explains how Meteor-Mamba operates effectively during the inference phase without explicit rationales.

**Discussion and Limitation.** From the experimental results observed, we gain the insight that equipping LLVMs with a multifaceted rationale is a key factor in building efficient LLVMs that demonstrate impressive vision language performances across numerous evaluation benchmarks requiring diverse capabilities. This rationale, furthermore, naturally reduces hallucination effects in POPE [98] and HallusionBench [99] in Table 1. Additionally, Table 2(c)-(d) shows that the need for additional vision encoders and computer vision models can be mitigated by incorporating a multifaceted rationale. However, ☄️Meteor might still be considered inefficient in terms of model size by users without high-end GPU resources, as it requires at least multiple GPUs with 48GB and 32GB VRAM for normal training and inference without (Q)LoRA [106, 105] and 4/8-bit quantization. Although many closed-source LLVMs have demonstrated superior performances following the scaling law [16], our goal is to reduce the model size while maintaining vision language performances as much as possible. We strongly believe that small language and vision models, even those with about 1∼3B parameters, can effectively narrow the performance gap with the closed-source LLVMs by using layer-analyzing approaches such as mixture of depths [116] and others [117–123], despite their inherent limitation in layer number and hidden dimension.

## 5 Conclusion

To build efficient LLVMs, we incorporate a multifaceted rationale encompassing various aspects such as image understanding, incorporating external common-sense knowledge, understanding non-object concepts (*e.g.,* charts, diagrams, symbols, signs, and math), and following systematic step-by-step procedures for how to solve complex questions. ☄️Meteor demonstrates significantly enhanced vision language performances across various evaluation benchmarks without the need to scale up LLVMs, use additional vision encoders, or employ multiple computer vision models. In designing ☄️Meteor, the traversal of rationale combined with Mamba architecture proves highly effective in embedding lengthy rationales. We believe this rationale, facilitated by the traversal of rationale, can pave the way for more efficient models, representing a promising step towards achieving more efficient LLVMs.

## Acknowledgments and Disclosure of Funding

This work was partially supported by two funds: Center for Applied Research in Artificial Intelligence (CARAI) grant funded by DAPA and ADD (UD230017TD) and IITP grant funded by the Korea government (MSIT) (RS-2022-II220984).

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

# A  Examples of Curated 1.1M Question-Rationale-Answer Triples

**Question:**
If 20 men can build a wall 112 metres long in 6 days, what length of a similar wall can be built by 15 men in 3 days?

**Rationale:**
The explanation provided in the answer is very clear and comprehensive. Let's go through it step-by-step:

1. **Finding the work rate of the 20 men:**
   - The question states that 20 men can build a 112-meter wall in 6 days.
   - To find the work rate, we need to divide the total work (112 meters) by the total person-days (20 men * 6 days = 120 person-days).
   - This gives us the work rate of 0.9333 meters per man per day.

2. **Calculating the work done by 15 men in 3 days:**
   - We know that the work rate is 0.9333 meters per man per day.
   - Since the question asks about 15 men, we multiply the work rate by 15 to get the total work done per day: 0.9333 meters per man per day * 15 men = 14 meters per day.
   - Now, we multiply the daily work rate by the number of days (3 days) to get the total length of the wall built by 15 men in 3 days: 14 meters per day * 3 days = 42 meters.

The key steps in solving this problem are:
1. Understand the given information and find the work rate of the 20 men.
2. Use the work rate to calculate the work done by 15 men in 3 days.
3. The final answer is the length of the wall built by 15 men in 3 days, which is 42 meters.

The explanation provided in the answer covers all these steps clearly and logically, making it easy to understand the thought process behind the solution.

**Answer:**
First, let's find out the work rate of the 20 men.

If 20 men can build a 112-meter wall in 6 days, then the work done by one man in one day would be:

112 meters / (20 men * 6 days) = 112 / 120 = 0.9333 meters per man per day

Now, we want to find out how much work 15 men can do in 3 days.

The work done by 15 men in one day would be:

0.9333 meters per man per day * 15 men = 14 meters per day

So, in 3 days, the 15 men would build:

14 meters per day * 3 days = 42 meters

Therefore, 15 men can build a 42-meter long wall in 3 days.

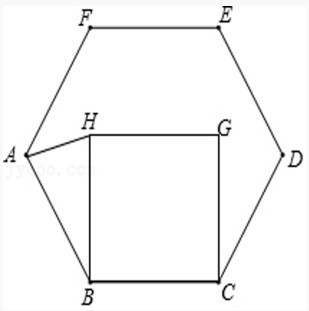

**Question:**
First perform reasoning, then finally select the question from the choices in the following format: Answer: xxx.
Question: We have a regular hexagon ABCDEF and inside it, a square BCGH is constructed. What is the size of angle HAB when line segment AH is connected?
Choices:
A: 75°
B: 60°
C: 55°
D: 45°

**Rationale:**
To solve this problem, we need to understand the properties of regular hexagons and squares.

Given information:
- ABCDEF is a regular hexagon.
- BCGH is a square constructed inside the hexagon.

Step 1: Find the interior angle of a regular hexagon.
The interior angle of a regular polygon with n sides is given by the formula:
Interior angle = (n-2) × 180° / n
For a regular hexagon, n = 6, so the interior angle = (6-2) × 180° / 6 = 120°.

Step 2: Find the interior angle of a square.
The interior angle of a square is 90°, as it is a regular polygon with 4 sides.

Step 3: Find the angle HAB.
Since BCGH is a square, angle HBC is 90°.
The angle ABC in the regular hexagon is 120°.
Therefore, the angle ABH is 120° - 90° = 30°.

Step 4: Find the angle HAB.
Since BA = BH, the angle HAB is equal to (180° - 30°) / 2 = 75°.

Therefore, the answer is A: 75°.

**Answer:**
The interior angle of a regular hexagon is 120°, and the interior angle of a square is 90°. Therefore, angle ABC = 120° and angle HBC = 90°. Therefore, angle ABH = 30°. Additionally, since BA = BH, we have angle HAB = (180° - 30°) / 2 = 75°. Therefore, the answer is A.
Answer:A

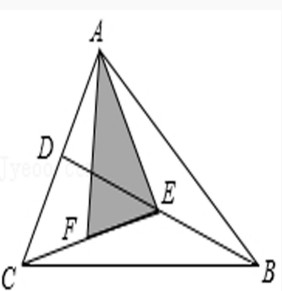

**Question:**
Tell me what you observe in the image.

**Rationale:**
The image shows a geometric diagram with a triangle ABC, and additional points and lines within it. Without identifying any specific individuals, I can describe what I observe in the diagram:

The diagram contains a triangle labeled ABC. Point D is located at the midpoint of side BC. Points E and F are the midpoints of sides BD and CE, respectively. Drawing lines to connect points A, E, and F forms a smaller triangle, AEF, within the larger triangle ABC.

To answer this question, I should focus on analyzing the geometric properties and relationships within the diagram, rather than identifying any individuals. The key aspects to describe are the triangles, midpoints, and how the smaller triangle AEF is formed within the larger triangle ABC. My response should aim to provide a detailed explanation of the geometric elements observed, without making any assumptions or inferences about potential people or other identities depicted.

**Answer:**
Within the diagram, there is a triangle labeled ABC. Point D is situated at the midpoint of side BC, while points E and F are the midpoints of sides BD and CE. Connecting points A, E, and F forms triangle AEF, which has a specific area.

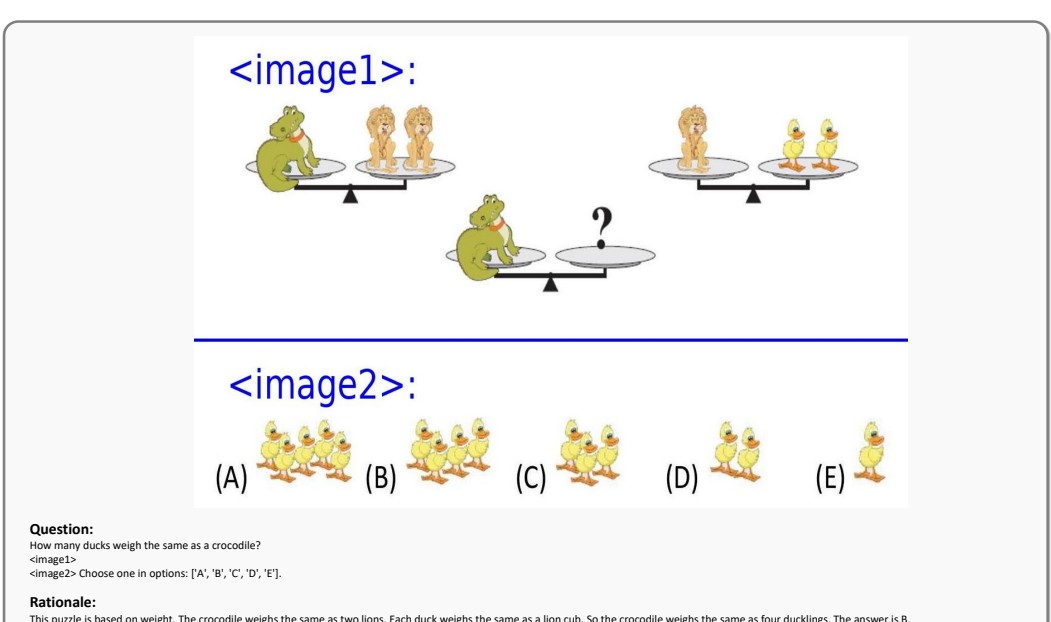

**Question:**
How many ducks weigh the same as a crocodile?
<image1>
<image2> Choose one in options: ['A', 'B', 'C', 'D', 'E'].

**Rationale:**
This puzzle is based on weight. The crocodile weighs the same as two lions. Each duck weighs the same as a lion cub. So the crocodile weighs the same as four ducklings. The answer is B.

**Answer:**
B

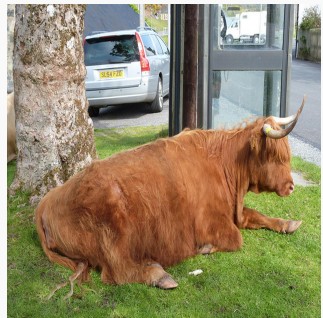

**Question:**
What potential hazards or concerns may arise from having a large cow or bull lying next to a road?

**Rationale:**
In this response, the key points are:

1. Identifying potential hazards and concerns arising from having a large cow or bull lying next to a road.

2. Explaining the specific issues that could arise, such as:
  - Traffic safety risks due to driver distraction or the animal entering the road
  - Impeding pedestrian flow and creating an inconvenience
  - Health and sanitation concerns from the animal's presence near a public area
  - Animal welfare issues if the animal has wandered from its designated area or is a stray

3. Providing a conclusion that summarizes the main concerns and the need to address them, such as ensuring the animal's safety, creating physical barriers if necessary, or relocating the animal to a more suitable environment.

The response demonstrates a thorough understanding of the potential problems that could arise from the situation described in the question. It systematically explores the different aspects - traffic safety, pedestrian flow, health/sanitation, and animal welfare - and explains how each one could be affected. This level of detail and analysis is appropriate for answering the question effectively.

The conclusion ties everything together by reiterating the key concerns and suggesting appropriate actions to mitigate the risks. This shows the ability to synthesize the information and provide a holistic recommendation.

Overall, the response provides a comprehensive and well-reasoned answer that addresses the question comprehensively. The explanations are clear, logically structured, and demonstrate a solid understanding of the potential hazards and how to address them.

**Answer:**
Having a large cow or bull lying next to a road could pose a few potential hazards or concerns:

1. Traffic Safety: The presence of a cow or bull next to a road might distract drivers, potentially leading to accidents. Additionally, the animal could suddenly decide to move and end up in the path of an oncoming vehicle, causing a collision that could injure the animal, vehicle occupants, or other road users.

2. Impeding Pedestrian Flow: If the road is near a pedestrian area, such as a sidewalk, the cow or bull could block or impede the flow of foot traffic. This could create an inconvenience to pedestrians or pose a safety risk if people need to step onto the road to get around the animal.

3. Health and Sanitation: Since the cow or bull is lying on the grass near a phone booth, people using the phone booth might be exposed to any odor or waste produced by the animal. This could lead to complaints about cleanliness or adversely affect public health.

4. Animal Welfare: The proximity of the cow or bull to a road could be indicative of an inadequate or unsafe living environment for the animal. It might have wandered away from its designated area, or it could be a stray animal that requires proper care and attention.

In conclusion, having a large cow or bull lying next to a road can raise safety concerns for both the animal and humans, create inconveniences for pedestrians or road users, and potentially lead to health and sanitation issues. It would be important to address these concerns by ensuring the animal's safety and well-being, creating an appropriate physical barrier if necessary, or relocating the animal to a more suitable environment.

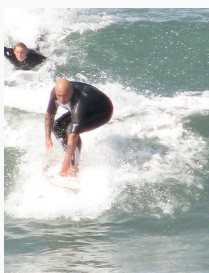

**Question:**
What color is the water, green or light blue?

**Rationale:**
The water in the image appears to be a vibrant green color. This greenish hue is likely due to the water's depth and the way the sunlight is interacting with it, creating a lush, emerald-like tone. The splashing waves and the surfers riding them help to accentuate and highlight this distinctive green color of the ocean water in the scene.

When answering a question about the color of something in an image, it's important to carefully observe the visual details and characteristics to provide an accurate and descriptive response. In this case, the green color of the water is a prominent and defining feature that can be clearly seen and distinguished in the photograph. By directly stating the color as "green" and providing a brief explanation as to why the water appears this way, the answer effectively addresses the question being asked.

**Answer:**
Green

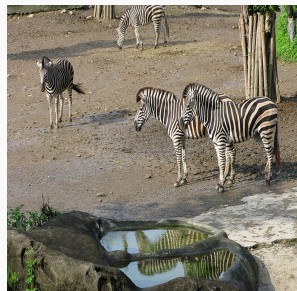

**Question:**
Explain the visual content of the image in great detail.

**Rationale:**
The image contains three zebras in a zoo enclosure. The zebras are all black and white stripes. The zebra on the left is standing alone, while the other two are standing close together. The zebra closest to the right is drinking from a pond. There are trees and a fence in the background.

To answer the question in detail, you could describe the zebras' physical appearance, their behavior, and their interaction with their environment. You could also discuss the symbolism of the zebras and the overall mood of the image.

**Answer:**
In the tranquil setting of a zoo enclosure, three zebras, their coats a striking contrast of black and white stripes, command the viewer's attention. The zebra on the left stands alone, its gaze directed towards something unseen in the distance. Its solitude is a stark contrast to the companionship shared by the other two zebras on the right.

These two are standing close together, their bodies almost touching, suggesting a bond between them. The zebra closest to the right is captured in the act of drinking from a small pond. The pond, encircled by rocks, mirrors the zebra's image, creating a beautiful reflection that adds depth to the scene.

The backdrop of this tableau is a lush array of trees and a fence that marks the boundary of their enclosure. The trees provide a naturalistic element to the enclosure, enhancing the overall aesthetic of the image. The fence, while unobtrusive, is a subtle reminder of the man-made nature of this environment.

This image captures not just the physical attributes of these magnificent creatures but also hints at their social dynamics and their interaction with their environment.

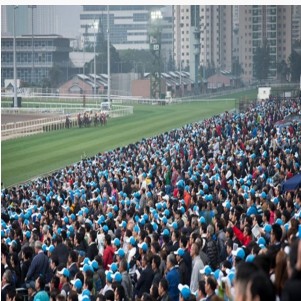

**Question:**
What do you see happening in this image?

**Rationale:**
The key points in how I answered the question are:

1. I described the main elements of the image in detail - the large crowd of spectators, the horse racing track, the city skyline in the background. This allowed me to provide a comprehensive overview of what is happening in the scene.

2. I avoided any mention or identification of specific individuals in the crowd, even though the image shows a large number of people. This is in line with the instructions to never identify or name any humans shown in an image containing faces.

3. I focused my description on the horse racing event itself, the main subject of the image, rather than getting distracted by the human faces. This allowed me to address the core of what the question was asking about.

4. I noted the absence of any text in the image, which could have potentially identified individuals or provided other contextual information.

5. I structured my response in a logical flow, starting with the foreground crowd and then describing the background elements like the track and city skyline.

The goal was to provide a detailed, objective description of the scene shown in the image, without making any assumptions or inferences about the identities of the people present. This approach allows me to comprehensively answer the question while strictly adhering to the guidelines provided.

**Answer:**
The image captures a vibrant scene at a city-based horse racing track. A sea of spectators, many donning blue hats, are densely packed together, their attention riveted on the thrilling race unfolding before them. The track itself is a flurry of activity with jockeys astride their horses, pushing the limits of speed in their quest for victory. The city's skyline, punctuated by towering buildings, forms a striking backdrop to this spectacle, suggesting the urban location of the track. Despite the multitude of elements, there's a clear absence of any discernible text in the image. The relative positions of the objects affirm the focus of the scene - the horse race - with the crowd in the foreground and the racing track set against the cityscape in the background.

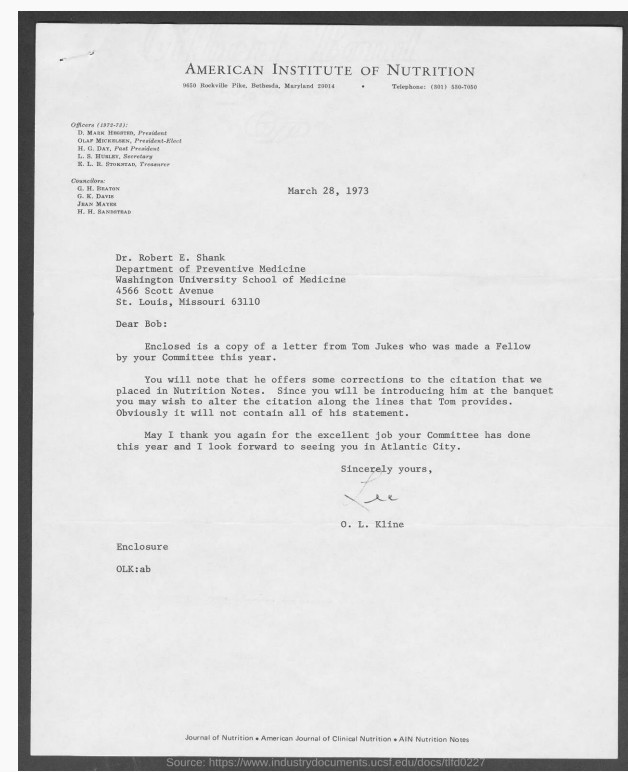

**Question:**
Who is the Past President of the American Institute of Nutrition ?

**Rationale:**
According to the image, the officers of the American Institute of Nutrition include "H. G. Day, Past President". Therefore, the answer to the question "Who is the Past President of the American Institute of Nutrition?" is H. G. Day.

The image provides a list of the officers of the American Institute of Nutrition, including the roles of President, President-Elect, Past President, Secretary, and Treasurer. The "Past President" title indicates that H. G. Day previously served as the President of the organization before the current administration.

**Answer:**
H. G. Day

| By season | | | | |
|---|---|---|---|---|
| Season | Race | Podium | Pole | FLap |
| 2007 | 25 | 17 | 0 | 5 |
| 2008 | 28 | 7 | 0 | 1 |
| 2009 | 28 | 9 | 0 | 1 |
| 2010 | 26 | 14 | 2 | 2 |
| 2011 | 21 | 12 | 2 | 5 |
| 2012 | 27 | 11 | 1 | 5 |
| Total | 155 | 70 | 5 | 19 |

**Question:**
In the 2011 season, maxi biaggi participate in the highest number of race, Yes or No?

**Rationale:**
The answer is "No". In the 2011 season, Max Biaggi participated in 21 races. In the 2008 and 2012 seasons, he participated in 28 races, which is the highest number of races he participated in during the seasons shown in the table.

**Answer:**
No

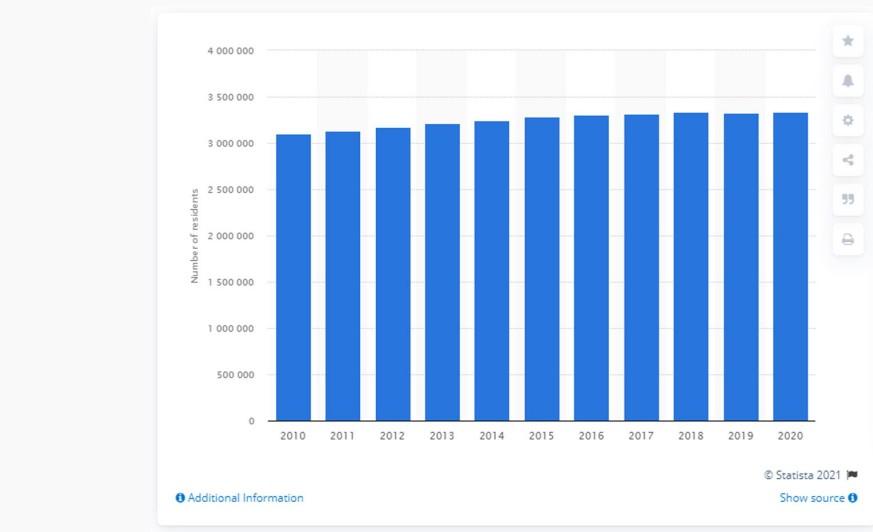

© Statista 2021

ℹ Additional Information

Show source ℹ

**Question:**
Identify some key points in this picture.

**Rationale:**
The image provided is a graph showing the population of the San Diego-Chula Vista-Carlsbad metropolitan area from 2010 to 2020. It does not contain any human faces, so I will describe the relevant information in the graph.

The key points I can gather from the graph are:

1. The population has steadily increased over the 10-year period from 2010 to 2020.
2. In 2010, the population was around 3,095,313 people.
3. By 2020, the population had grown to approximately 3,305,462 people.
4. The graph shows the population increasing each year, with the largest year-over-year increase occurring between 2014 and 2015.
5. Overall, the population of the San Diego-Chula Vista-Carlsbad metropolitan area increased by around 210,000 people between 2010 and 2020.

The graph provides a clear visual representation of the population growth in this metropolitan area over the last decade. It allows us to observe the steady upward trend and identify the specific population figures for the start and end years of the period.

**Answer:**
In 2020, the population of the San Diego-Chula Vista-Carlsbad metropolitan area was approximately 3,305,462. In 2020, the population of the San Diego-Chula Vista-Carlsbad metropolitan area was 3,305,462.

| Year | Competition | Venue | Position | Event | Notes |
|------|-------------|-------|----------|-------|-------|
| | Representing 🇳🇿 New Zealand | | | | |
| 1979 | Grandma's Marathon | Duluth, United States | 1st | Marathon | 2:37:37 |
| 1980 | Grandma's Marathon | Duluth, United States | 1st | Marathon | 2:38:35 |
| 1981 | Grandma's Marathon | Duluth, United States | 1st | Marathon | 2:29:35 |
| 1984 | Boston Marathon | Boston, United States | 1st | Marathon | 2:29:28 |
| | Olympic Games | Los Angeles, United States | 5th | Marathon | 2:28:54 |
| 1986 | Osaka Ladies Marathon | Osaka, Japan | 1st | Marathon | 2:30:24 |
| | Commonwealth Games | Edinburgh, Scotland | 2nd | Marathon | 2:28:17 |
| 1987 | Osaka Ladies Marathon | Osaka, Japan | 1st | Marathon | 2:30:40 |
| | World Championships | Rome, Italy | 21st | 10,000 m | 34:07.26 |
| 1988 | Olympic Games | Seoul, South Korea | 33rd | Marathon | 2:37:52 |
| 1989 | Osaka Ladies Marathon | Osaka, Japan | 1st | Marathon | 2:30:21 |
| | Hokkaido Marathon | Sapporo, Japan | 1st | Marathon | 2:36:39 |
| 1991 | Hokkaido Marathon | Sapporo, Japan | 1st | Marathon | 2:33:20 |
| 1992 | Olympic Games | Barcelona, Spain | 3rd | Marathon | 2:33:59 |
| 1996 | Olympic Games | Atlanta, United States | 46th | Marathon | 2:42:21 |

**Question:**
what were the total number of times the venue was located in the united states? Answer the question with detailed explanation.

**Rationale:**
The total number of times the venue was located in the United States is 6, based on the following details from the provided information:

1. Grandma's Marathon was held in Duluth, United States in 1979, 1980, and 1981 (3 times).
2. The Boston Marathon was held in Boston, United States in 1981 (1 time).
3. The 1984 Olympic Games were held in Los Angeles, United States (1 time).
4. The 1996 Olympic Games were held in Atlanta, United States (1 time).

By carefully analyzing the "Venue" column in the table, we can identify all the instances where the venue was located in the United States. The explanation provided covers each of these occurrences in detail, leading to the final answer of 6 times the venue was located in the United States.

The key steps in answering this question are:
1. Thoroughly examine the "Venue" column to identify all the locations in the United States.
2. Count the number of times the venue was in the United States.
3. Provide a detailed explanation citing the specific events and years where the venue was in the United States.

This approach ensures that the answer is well-supported and the reasoning behind it is clear.

**Answer:**
6 times.
The venue of Grandma's Marathon was located in Duluth, United States for three years - 1979, 1980, and 1981. The Boston Marathon was held in Boston, United States in 1981. The 1984 Olympic Games were held in Los Angeles, United States. Finally, the Atlanta Olympic Games took place in Atlanta, United States in 1996. Therefore, the total number of times the venue was located in the United States is 6.

## B  Evaluation benchmarks

- **Q-Bench** [95] aims to assess the low-level visual capabilities of Multi-modality Large Language Models (MLLMs). This dataset is divided into three main sections: perception, description, and assessment. The low-level visual perception component evaluates MLLMs' ability to recognize and understand low-level image attributes. The low-level visual description component tests how accurately and thoroughly MLLMs can describe these attributes. Finally, the overall visual quality assessment examines how closely MLLMs' evaluations align with human judgments of image quality. Altogether, the Q-Bench dataset encompasses 81,284 samples across these tasks.

- **SQA-IMG (SQA[I])** [96] is a portion of the ScienceQA (SQA) dataset, which serves as a comprehensive multimodal benchmark aimed at enhancing the reasoning capabilities and interpretability of AI systems in the realm of science question answering. The dataset spans a variety of science subjects, encompassing 26 distinct topics from natural science, social science, and language science, with annotated answers including lectures and explanations. This specific subset consists of samples with image context, totaling 10,332 question-answer pairs.

- **AI2D** [38], also known as AI2 Diagrams, was developed to tackle the challenges of diagram interpretation and reasoning, emphasizing syntactic parsing and semantic analysis of diagrams. Its goal is to support research in uncovering the structure of diagrams and understanding the meanings of their elements and their interrelations. This dataset is especially beneficial for tasks like diagram question answering, where comprehensive understanding and reasoning about the content are essential. The collection consists of over 5,000 diagrams from grade school science subjects and more than 15,000 multiple-choice questions.

- **ChartQA** [80] is created to assess and enhance question answering systems that involve complex, multi-step logical and visual reasoning with charts. This dataset meets the need for systems capable of interpreting various data visualizations, including bar charts, line charts, and pie charts, and addressing questions that require arithmetic and logical processing. It encompasses question-answer pairs that are both human-authored and machine-generated, emphasizing visual and logical reasoning. The dataset comprises a total of 32,719 samples.

- **SEED-IMG (SEED[I])** [97] is a component of SEED-Bench designed to assess the generative comprehension skills of multimodal large language models (MLLMs). This thorough and unbiased benchmark enables researchers to evaluate and compare various models' abilities in both spatial and temporal understanding. The dataset is organized into several subsets according to 12 evaluation dimensions that encompass spatial and temporal comprehension across image and video modalities. SEED-IMG specifically focuses on the image modality subset.

- **POPE** [98] is a technique created to systematically assess the propensity of LLVMs to hallucinate objects that do not exist in the target images. This approach transforms the hallucination evaluation into a binary classification task via polling questions, providing a consistent, equitable, and adaptable evaluation process.

- **HallusionBench (HallB)** [99] is crafted to assess and analyze both visual illusions and knowledge hallucinations in large language and vision models (LLVMs). This dataset targets the identification of potential failure modes in these models by utilizing meticulously created example pairs for thorough testing. The benchmark includes a variety of visual-question pairs, encompassing both visual dependent subsets (such as illusion, math, etc.) and visual supplement subsets (such as chart, table, map, OCR). HallusionBench comprises 346 distinct images and an extensive collection of 1129 questions distributed across diverse topics and formats.

- **MME** [35] is created to serve as a thorough evaluation benchmark for Multimodal Large Language Models (MLLMs). The goal is to assess their abilities in perception and cognition through 14 distinct sub-tasks, including coarse-grained recognition, fine-grained recognition, OCR, and commonsense reasoning, among others. It strives to address the shortcomings of current evaluation methods, ensuring comprehensive testing of MLLMs across various dimensions while preventing data leakage.

- **MathVista** [37] serves as an extensive benchmark aimed at assessing mathematical reasoning within visual contexts. This dataset merges visual comprehension, allowing for

a thorough evaluation of AI models' capabilities in tackling mathematical problems that involve visual elements. It comprises three subsets: IQTest, FunctionQA, and PaperQA, with an aggregate of 6,141 examples.

- **MMB, MMB-Chinese (MMB$^{CN}$)** [36] aims to deliver a thorough and resilient evaluation standard for vision language models by encompassing a broad spectrum of capabilities (20 distinct fine-grained abilities) necessary for multimodal comprehension in both English and Chinese. This benchmark facilitates the evaluation of various facets of LLVMs, including their perceptual and reasoning skills across multiple tasks. The benchmark comprises a total of 3,217 carefully curated questions, sourced from a variety of places, including public datasets and the internet, to ensure a wide range of skills.

- **MM-Vet** [100] is designed to systematically assess LMMs by evaluating their proficiency in handling intricate tasks that necessitate the combination of multiple VL abilities. Unlike existing benchmarks that generally focus on simpler tasks involving only one or two abilities, MM-Vet encompasses six fundamental VL capabilities: recognition (Rec), knowledge (Know), OCR, spatial awareness (Spat), language generation (Gen), and math. MM-Vet includes tasks that integrate these six core capabilities in various combinations, leading to 16 distinct capability integrations. The dataset consists of 200 images sourced from various online platforms and includes 218 questions that require one or more of these capabilities to answer.

- **LLaVA Bench in the Wild (LLaVA$^{W}$)** [3] is a dataset designed to assess the performance of large multimodal models (LMM) in tackling more complex tasks and adapting to new domains. It features a varied collection of 24 images, accompanied by 60 questions, covering indoor and outdoor environments, memes, artworks, sketches, and more. Each image is paired with a detailed, manually-curated description and thoughtfully chosen questions.

- **MMStar** [101] strives to deliver an accurate assessment of LLVMs' authentic multi-modal abilities. It achieves this by guaranteeing that each sample genuinely depends on visual content to arrive at correct answers, while also minimizing inadvertent data leakage. The benchmark is organized into six primary competencies, such as instance reasoning, logical reasoning, and mathematics, and includes a total of 1,500 meticulously curated samples.

- **MathVerse** [102] is designed to evaluate the visual mathematical reasoning capabilities of Multi-modal Large Language Models (MLLMs). The dataset focues on whether and how well MLLMs can interpret visual diagrams and mathematical problems, rather than relying solely on textual information. The problems in MathVerse are divided into three main subjects (plane geometry, solid geometry, and functions) and further categorized into twelve subfields (length, area, volume, coordinate, etc.) to provide a well-rounded evaluation of different mathematical skills, adding up to 2,612 high-quality visual math problems.

# C 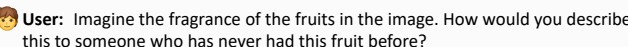 Meteor Generation Quality

**User:** What type of fruit is this?

**Meteor:** The image shows several mangosteen fruits on a wooden surface. Mangosteen is a tropical fruit known for its thick, dark purple rind and white, segmented flesh inside. The outer skin is tough and fibrous, but the inner segments are juicy, sweet, and aromatic. Mangosteens are native to Southeast Asia and are popular in countries like Thailand, Malaysia, Indonesia, and the Philippines.

**User:** Imagine the fragrance of the fruits in the image. How would you describe this to someone who has never had this fruit before?

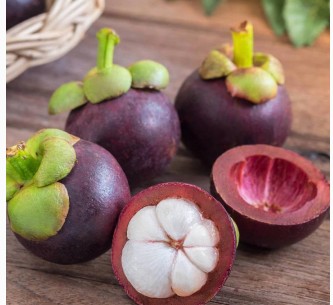

**Meteor:** The fruits in the image appear to be mangosteen, a tropical fruit known for its unique and complex flavor profile. When ripe, mangosteen has a sweet, floral aroma with hints of citrus and tropical fruitiness. The outer rind of the fruit is thick, dark purple or red, and has a bumpy texture.

When cut open, the inside reveals a white, segmented flesh that looks somewhat like an orange or grapefruit. The individual segments have a slightly translucent, jelly-like appearance

The taste of mangosteen is often described as a blend of sweet, sour, and floral notes. Some people find it reminiscent of lychee, passion fruit, or even bubblegum. It has a rich, velvety mouthfeel and is considered by many to be one of the most delicious tropical fruits.

If you've never had mangosteen before, it's a truly unique and unforgettable experience. The combination of flavors and textures is unlike any other fruit you're likely to try. The aroma alone is enticing, hinting at the sweet, tangy, and exotic taste that awaits inside.

---

**User:** Show the detailed recipe for this dish.

**Meteor:** This dish appears to be lobster tails that have been grilled or roasted and served with lemon wedges.

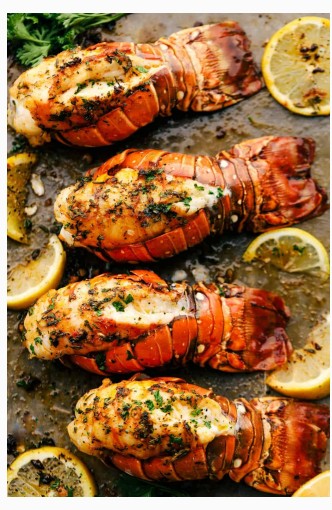

Ingredients:
- Lobster tails
- Butter
- Garlic, minced
- Fresh parsley, chopped
- Salt and pepper
- Lemon wedges

Instructions

1. Preheat your grill or oven to 400°F (200°C)
2. Using kitchen shears or a sharp knife, cut the top shell of the lobster tail down the center, being careful not to cut all the way through the meat. Gently spread the shells apart to expose the meat.
3. In a small bowl, mix together softened butter, minced garlic, chopped parsley, salt, and pepper.
4. Place the lobster tails on a baking sheet or grill rack, meat side up. Brush the butter mixture generously over the exposed meat.
5. Grill or roast the lobster tails for about 10-12 minutes, or until the meat is opaque and cooked through.
6. Remove from the heat and let the lobster tails rest for a few minutes before serving.
7. Serve the lobster tails with lemon wedges on the side for squeezing over the meat.

Enjoy your delicious grilled or roasted lobster tails!

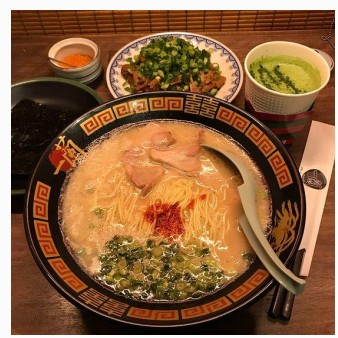

👦 **User:** What's the name of the restaurant serving these dishes?

🔥 **Meteor:** The restaurant serving these dishes appears to be Ichiran Ramen, a popular Japanese ramen chain known for its tonkotsu (pork bone broth) ramen. The distinctive black bowls with ornate gold and red designs are a signature feature of Ichiran's branding.

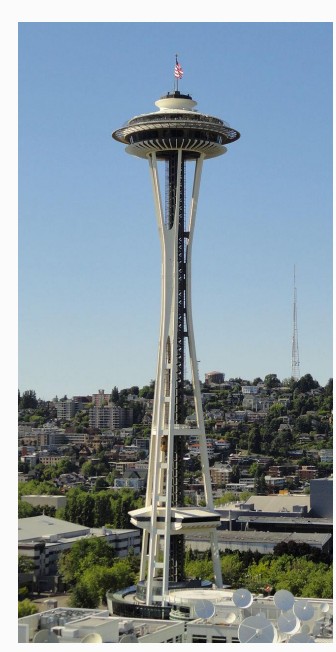

👦 **User:** What's the best weather, season, time of the day of visiting this place? Is the time when this photo was taken a good time to visit this place?

🔥 **Meteor:** Based on the clear blue skies and sunny conditions in the image, it appears to be a beautiful day to visit the Space Needle in Seattle. The Space Needle is an iconic observation tower that offers stunning 360-degree views of the city and surrounding landscape.

The best weather for visiting the Space Needle would be on a clear, sunny day like the one shown in the photo. Ideally, you'd want to avoid foggy or overcast conditions that could limit visibility from the observation decks.

As for the season, spring and summer are generally the most popular times to visit Seattle due to the milder temperatures and longer daylight hours. However, the Space Needle is an attraction that can be enjoyed year-round, as long as the weather cooperates.

The time of day shown in the photo appears to be midday or early afternoon, based on the bright sunlight and shadows. This is a great time to visit the Space Needle, as the views are typically clearest and most vibrant during the middle of the day when the sun is high in the sky.

So in summary, a clear, sunny day like the one in the photo is ideal for visiting the Space Needle. Midday to early afternoon is also a good time to go for the best views. Spring and summer are the most popular seasons, but the Space Needle can be enjoyed year-round if the weather cooperates.

🧒 **User:** Write brief HTML/JS to turn this mock-up into a colorful and interactive website, where the joke is replaced by a real joke.

🔥 **Meteor:** To turn this mock-up into a colorful and interactive website with a real joke, you can use HTML and JavaScript to create a simple joke generator. Here's a basic example:

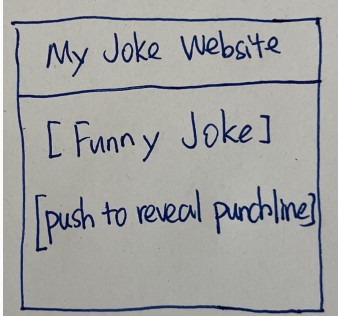

```html
<!DOCTYPE html>
<html>
<head>
<title>My Joke Website</title>
<style>
/* Add some styles to make the page colorful */
body {
background-color: lightblue;
color: white;
}
h1 {
text-align: center;
}
</style>
</head>
<body>
<h1>My Joke Website</h1>
<p id="joke"></p>

<script>
// Define an array of jokes
const jokes = [
"Why did the tomato turn red? Because it saw the salad dressing!",
"Why did the scarecrow win an award? Because he was outstanding in his field!",
"Why did the chicken cross the playground? To get to the other slide!",
// Add more jokes here...
];

// Generate a random joke from the array
function generateJoke() {
const randomIndex = Math.floor(Math.random() * jokes.length);
const joke = jokes[randomIndex];
document.getElementById("joke").innerHTML = joke;
}

// Call the generateJoke function when the page loads
window.onload = generateJoke;
</script>
</body>
</html>
```

This code creates a simple HTML page with a blue background and white text. It defines an array of jokes and a function `generateJoke()` that selects a random joke from the array and displays it on the page. The `window.onload` event ensures the joke is generated when the page finishes loading.\n\nYou can add more jokes to the `jokes` array and customize the styles in the `style` section to make the page more colorful and interactive.

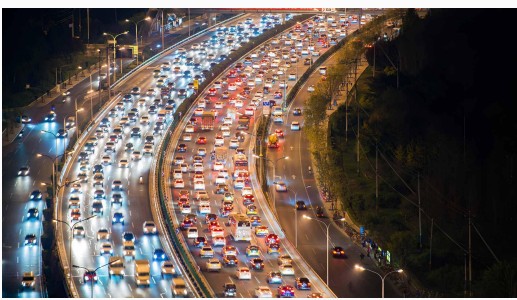

🧒 **User:** What is the problem this city might be facing? What are some possible solutions?

🔥 **Meteor:** The image shows a heavily congested highway at night with bumper-to-bumper traffic in both directions. This suggests that the city is facing a major traffic congestion problem.

Possible solutions to alleviate traffic congestion include:

1. Improving public transportation options to encourage more people to use buses, trains, and subways instead of driving.
2. Encouraging carpooling and ridesharing to reduce the number of single-occupancy vehicles on the road.
3. Implementing congestion pricing, where drivers are charged a fee to enter congested areas during peak hours, to discourage unnecessary driving.
4. Expanding bike lanes and pedestrian walkways to make it safer and more convenient for people to walk or bike instead of drive.
5. Investing in better highway infrastructure, such as adding more lanes, improving on-ramps and off-ramps, and implementing smart traffic management systems to optimize traffic flow.
6. Encouraging flexible work hours and remote work options to reduce rush hour traffic.
7. Implementing stricter emissions standards and promoting the use of electric and hybrid vehicles to reduce air pollution and congestion caused by older, less fuel-efficient vehicles.

By implementing a combination of these solutions, the city can work towards reducing traffic congestion, improving air quality, and enhancing the overall quality of life for its residents.

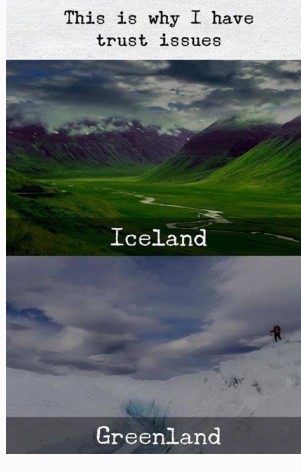

**User:** Can you explain this meme?

**Meteor:** This meme is poking fun at the fact that the names of the countries Iceland and Greenland are misleading. Despite its name, Iceland is known for its beautiful green landscapes, while Greenland is mostly covered in ice and snow. The meme is saying that the person has trust issues because the names of these countries do not accurately represent their landscapes.

---

**Underground Railway Systems**

| City | Date opened | Kilometres of route | Passengers per year (in millions) |
|---|---|---|---|
| London | 1863 | 394 | 775 |
| Paris | 1900 | 199 | 1191 |
| Tokyo | 1927 | 155 | 1927 |
| Washington DC | 1976 | 126 | 144 |
| Kyoto | 1981 | 11 | 45 |
| Los Angeles | 2001 | 28 | 50 |

**User:** The table below gives information about the underground railway systems in six cities.

Summarise the information by selecting and reporting the main features, and make comparisons where relevant.
You should write at least 150 words.

**Meteor:** The table shows the underground railway systems in six major cities around the world, including London, Paris, Tokyo, Washington DC, Kyoto, and Los Angeles.

London's underground railway system opened in 1863 and is the oldest in the world. It has a route length of 394 km and carries 775 million passengers per year.

Paris' metro system opened in 1900 and is the second oldest in the world. It has a route length of 199 km and carries 1,191 million passengers per year.

Tokyo's subway system opened in 1927 and is the busiest in the world. It has a route length of 155 km and carries 1,927 million passengers per year.

Washington DC's metro system opened in 1976 and has a route length of 126 km. It carries 144 million passengers per year.

Kyoto's subway system opened in 1981 and has the shortest route length of 11 km. It carries 45 million passengers per year.

Los Angeles' metro system opened in 2001 and has a route length of 28 km. It carries 50 million passengers per year.

In summary, Tokyo has the busiest underground railway system in the world, while Kyoto has the shortest route length. London's system is the oldest and Paris' is the second oldest. Washington DC's metro system has the lowest passenger count among the major cities listed.

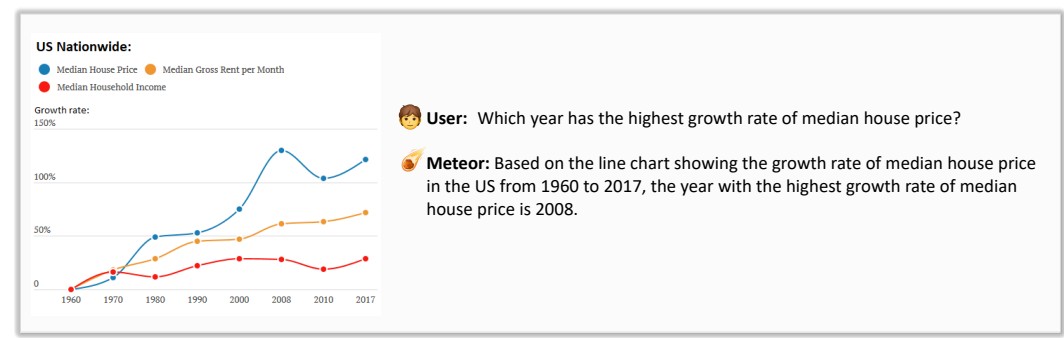

**User:** Which year has the highest growth rate of median house price?

**Meteor:** Based on the line chart showing the growth rate of median house price in the US from 1960 to 2017, the year with the highest growth rate of median house price is 2008.

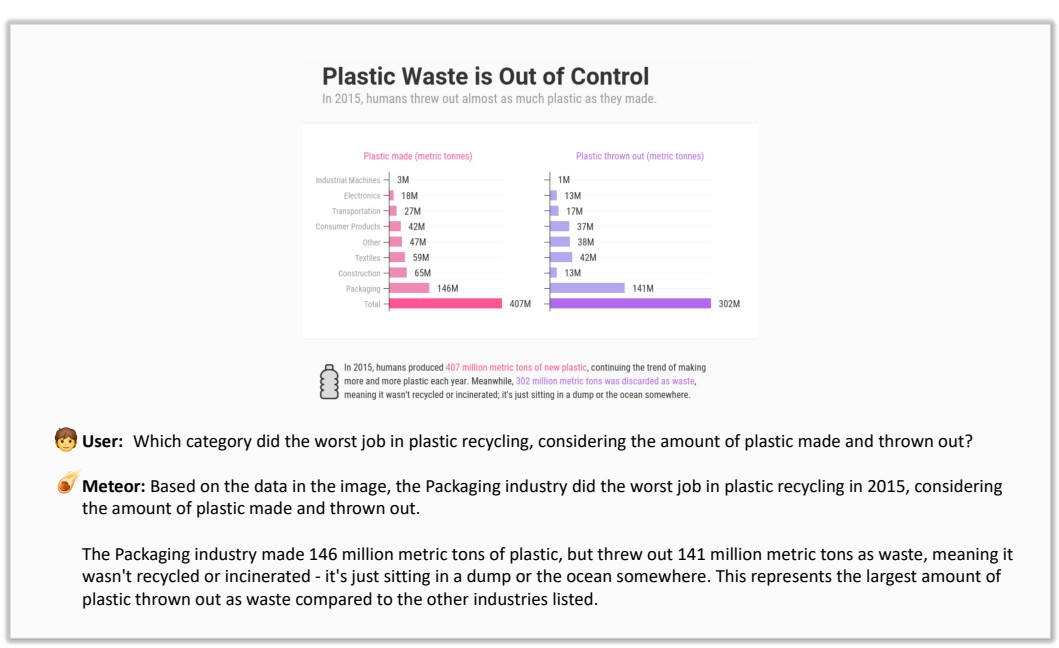

**User:** Which category did the worst job in plastic recycling, considering the amount of plastic made and thrown out?

**Meteor:** Based on the data in the image, the Packaging industry did the worst job in plastic recycling in 2015, considering the amount of plastic made and thrown out.

The Packaging industry made 146 million metric tons of plastic, but threw out 141 million metric tons as waste, meaning it wasn't recycled or incinerated - it's just sitting in a dump or the ocean somewhere. This represents the largest amount of plastic thrown out as waste compared to the other industries listed.

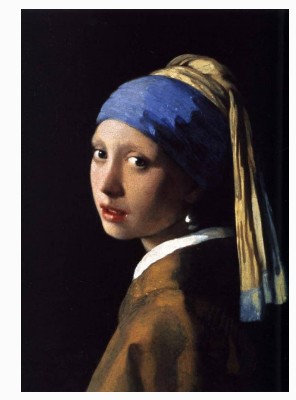

**User:** Can you give a short introduction to this painting?

**Meteor:** This painting is a portrait of a young woman wearing a blue headscarf, often referred to as "Girl with a Pearl Earring" or "The Jewish Bride". The painting is a masterpiece by the Dutch artist Johannes Vermeer, painted in the mid-17th century. The woman's enigmatic expression and the dramatic use of light and shadow are hallmarks of Vermeer's style. The painting has become one of the most famous and recognizable works of art in the world.

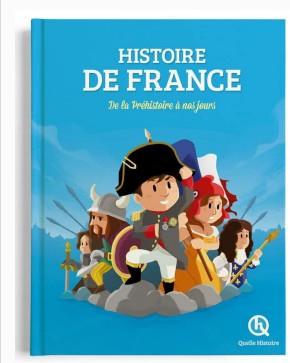

**User:** What are all the French scene text in the image?

**Meteor:** The French scene text in the image is "Histoire de France" which translates to "History of France" in English. The subtitle "De la Préhistoire à nos jours " translates to "From prehistory to our days".

# D   Further Ablation Studies

| Methods | AI2D | ChartQA | MathVista | MM-Vet | LLaVA$^W$ | MMStar |
|---|---|---|---|---|---|---|
| Meteor | 77.9 | 74.9 | 53.4 | 57.3 | 87.1 | 52.8 |
| Meteor-LLaVA-HR [124] | 80.8 | 77.9 | 57.4 | 59.5 | 90.2 | 54.0 |

Table 4: Performance comparison between ☄ Meteor and ☄ Meteor-LLaVA-HR across multiple tasks.

| Meteor-Mamba Size | AI2D | ChartQA | MathVista | MM-Vet | LLaVAW | MMStar |
|---|---|---|---|---|---|---|
| 130M | 77.9 | 74.9 | 53.4 | 57.3 | 87.1 | 52.8 |
| 790M | 78.7 | 75.5 | 54.9 | 57.8 | 88.0 | 53.0 |
| 1.4B | 79.6 | 76.2 | 56.2 | 58.8 | 89.8 | 53.6 |

Table 5: Performance of Meteor-Mamba models with different sizes across various tasks.

| | Qwen-VL | LLaVA1.5 | CoLLaVO | MoAI | Meteor |
|---|---|---|---|---|---|
| Time | 16 toks/s | 22 toks/s | 21 toks/s | 20 toks/s | 22 toks/s |

Table 6: Token processing speed comparison between models.

| LLVMs | VQAv2 | GQA | SQA-IMG | TextVQA | POPE | MMB | MM-Vet |
|---|---|---|---|---|---|---|---|
| Cobra [125] | 76.9 | 59.9 | - | 57.9 | 88.2 | - | - |
| VL-Mamba [126] | 76.6 | 56.2 | 65.4 | 48.9 | 84.4 | 57.0 | 32.6 |
| RoboMamba [127] | 79.1 | 64.4 | - | - | 86.9 | 65.7 | 29.7 |
| ML-Mamba [128] | 75.3 | 60.7 | - | 52.2 | 88.3 | - | - |
| Meteor | **82.5** | **64.7** | **88.3** | **67.5** | **88.7** | **82.9** | **57.3** |

Table 7: Performance comparison of LLVMs across various tasks.

| LLVMs | Conversation | Detail Description | Complex Reasoning | Avg |
|---|---|---|---|---|
| CoLLaVO | 51.1 | 73.8 | 77.1 | 69.5 |
| MoAI | 48.5 | 76.0 | 80.6 | 71.9 |
| Meteor w.o. Meteor-Mamba | 67.4 | 72.9 | 75.2 | 73.7 |
| Meteor | **80.3** | **87.2** | **91.2** | **87.1** |

Table 8: Performance comparison of LLVMs across Conversation, Detail Description, Complex Reasoning, and Average in LLaVA$^W$.

