# OpenReview forum: "Meteor: Mamba-based Traversal of Rationale for Large Language and Vision Models"
_NeurIPS.cc/2024/Conference — NeurIPS 2024 poster_

### Official Review · Reviewer_zsWf · 2024-07-07

**Soundness:** 3
**Presentation:** 3
**Contribution:** 3
**Rating:** 5
**Confidence:** 5

**Summary:**

This paper presents a Mamba-based traversal of rationales (Meteor) that leverages detailed image captions (multifaceted information) to provide more comprehensive image-related information to LLVMs, thereby enhancing the model's understanding and answering capabilities. The introduction of the Mamba architecture enables linear time complexity data processing, reducing the time required to generate multifaceted information from images.

**Strengths:**

This paper proposes a new approach to improve LLVM performance by enhancing image captions (model-based) without introducing additional vision encoders.

**Weaknesses:**

1. The introduction of Meteor-Mamba increases the model size, which contradicts the abstract. Additionally, Meteor's inference cost is higher than models like LLAVA and InternLM-XComposer2.
2. In Figure 3, what is the Meteor-Multimodal Language Model in the Second Training Step? Was it initialized with pre-trained MLLMs weights?
3. How does the Mamba architecture benefit Meteor? The experiments do not demonstrate the advantages of the Mamba structure.
4. I am puzzled by the performance achieved with only 2.1M instruction tuning. Please provide a detailed description of the training process, the data used, and the weights of each model employed.

**Questions:**

Same as Weaknesses

**Limitations:**

This paper does not reflect any potential negative societal impact.

---

> ### Author Rebuttal · Authors · 2024-07-31
>
> We appreciate your valuable comments! In the following rebuttals, we address all the comments you pointed out. We will definitely include these clarifications in our manuscript to enhance understanding in the potential camera-ready version.
>
> ---
>
> **Q1. The introduction of Meteor-Mamba increases the model size, which contradicts the abstract. Additionally, Meteor's inference cost is higher than models like LLAVA and InternLM-XComposer2.**
>
> **A1.** We believe the Meteor-Mamba 130M size is a marginal increase compared with the large language model size of 7B. Nonetheless, we will modify the expression to "without 'significantly' increasing model size" to clarify this point.
>
> Regarding inference time, please refer to Answer 1 of "Reviewer Hj2J," which describes the comparison of inference times. Based on this table, we can say that Meteor's inference cost is not higher than commonly used LLVMs.
>
> ---
>
> **Q2. In Figure 3, what is the Meteor-Multimodal Language Model in the Second Training Step? Was it initialized with pre-trained MLLMs weights?**
>
> **A2.** Yes, the pretrained language model is used in the second training step. The term "Meteor-Multimodal Language Model" in Figure 3 indicates that as training progresses, the pretrained language model evolves into a multimodal language model. We will add an explanation to clarify this in the manuscript.
>
> ---
>
> **Q3. How does the Mamba architecture benefit Meteor? The experiments do not demonstrate the advantages of the Mamba structure.**
>
> **A3.** For detailed reasoning capabilities of Meteor, please refer to Answer 2 tables of "Reviewer XZUR," which describes Meteor's complex reasoning capabilities on challenging benchmarks such as MM-Vet and LLaVA-W. In these tables, we showed the performance differences between with or without Meteor-Mamba for complex reasoning capabilities.
>
> ---
>
> **Q4. I am puzzled by the performance achieved with only 2.1M instruction tuning. Please provide a detailed description of the training process, the data used, and the weights of each model employed.**
>
> **A4.** Below are the details of the training process, datasets used, and models employed in each training step:
>
> 1. Training Process:
> > In the first training step, we only train Meteor-Mamba, the visual projector, and the tor projector. We autoregressively train only the partial segments of multifaceted rationales between special tokens $<tor>$.
> > In the second training step, we train all parameters except the vision encoder. Only tor tokens are propagated from Meteor-Mamba into Meteor-MLM without explicit rationale. We autoregressively train the answer parts for the given question prompts.
>
> 2. Used Dataset
> > We used a curated dataset of 1,059,382 (1.1M) Q-R-A triplet samples.
>
> The dataset breakdown is as follows:
>
> ```shell
> --------------------------------------------
> Real-World Image: 338K
> Document & Chart & Diagram & Sign & Symbol: 379K
> Math: 342K
>      Math with Vision: 165K
>      Math with Text only: 177K
> --------------------------------------------
>
> - ShareGPT4V-Caption (72507, 73K)
> - ShareGPT4V-Instruction (266072, 266K)
> - MiniGemini-Instruction (26885, 27K)
> - DocDownstream (298748, 299K)
> - DocReason (53065, 53K)
> - GLLaVA (162378, 162K)
> - MathVision (2992, 3K)
> - MathInstruct (81496, 81K)
> - MathPlus (95239, 95K)
> ```
>
> 3. Model employed in Each Training Step
> - Vision Encoder: CLIP-L/14
> - Pretrained Large Language Model: InternLM2-7B
> > In the first training step, we freeze the vision encoder and the pretrained large language model.
> > In the second training step, we freeze only the vision encoder.
>
> ---
>
> **We hope this rebuttal can improve the "Rating" score from Negatvie to Positive.**

---

> > ### Comment · Reviewer_zsWf · 2024-08-13
> >
> > Thanks the author for addressing my concerns. I would like to improve my score to borderline accept.

---

### Official Review · Reviewer_Hj2J · 2024-07-11

**Soundness:** 3
**Presentation:** 3
**Contribution:** 3
**Rating:** 7
**Confidence:** 5

**Summary:**

The paper proposes Meteor, which divides MLLM into two stages, i.e., rationale generation and question answering.  Experimental results show its effectiveness agasint existing MLLMs on many benchmarks.

**Strengths:**

1. The proposed idea is very reasonable. The technical implementation is also novel and strongly matches with the motivation.
2. The experiments are sufficient for me to understand the effectiveness of each component. Besides, the experimental results are good, which outperform most existing MLLMs.
3. I appreciate the design of the <tor> token, which bridges the gap between two LLMs in a simple and clear way.

**Weaknesses:**

1. I must admit the novel design of Metor, which however, also leads some pratical issues:
    How about the inference time against common MLLMs?
    Does this structure can be scaled up?
    Will Metor be better when combining high-resolution image encoders, e.g., LLaVA-HR[A].
    [A] Feast Your Eyes: Mixture-of-Resolution Adaptation for Multimodal Large Language Models
2. To be honest, the writing of the paper should be improved. I take a lot of time to understand the technical details. The method should be better oragnized.

**Questions:**

My main cocerns are the efficiency and generalization ability.

**Limitations:**

See weakness

---

> ### Author Rebuttal · Authors · 2024-07-31
>
> We appreciate your valuable comments! In the following rebuttals, we clarify the points you raised. We will incorporate this content into our manuscript in the next potential camera-ready stage.
>
> ---
>
> **Q1. How about the inference time against common MLLMs?**
>
> **A1.** We evaluated the inference time under equal resource environments: Intel(R) Xeon(R) Gold 6230, RAM 512GB, and NVIDIA RTX A6000, with flash attention applied. Despite Meteor using additional Meteor-Mamba module, it did not slow down the inference speed. This is because once Meteor-Mamba's embedded rationale features are acquired, no additional propagation of Meteor-Mamba is needed. Only the autoregressive decoding process in Meteor-MLM is required.
>
> |       |  Qwen-VL | LLaVA1.5 |  CoLLaVO |   MoAI   |  Meteor  |
> |:-----:|:--------:|:--------:|:--------:|:--------:|:--------:|
> |  Time | 16toks/s | 22toks/s | 21toks/s | 20toks/s | 22toks/s |
>
> ---
>
> **Q2. Does this structure can be scaled up? Will Meteor be better when combining high-resolution image encoders, e.g., LLaVA-HR**
>
> **A2.** The table below shows the results of scaling up the model size of Meteor-Mamba.
>
> | Meteor-Mamba Size | AI2D | ChartQA | MathVista | MM-Vet | LLaVAW | MMStar |
> |-------------------|:----:|:-------:|:---------:|:------:|:------:|:------:|
> | 130M              | 77.9 |   74.9  |    53.4   |  57.3  |  87.1  |  52.8  |
> | 790M              | 78.7 |   75.5  |    54.9   |  57.8  |  88.0  |  53.0  |
> | 1.4B              | 79.6 |   76.2  |    56.2   |  58.8  |  89.8  |  53.6  |
>
> We also applied Meteor to LLaVA-HR to adapt to high image resolution.
>
> |                 | AI2D | ChartQA | MathVista | MM-Vet | LLaVAW | MMStar |
> |-----------------|:----:|:-------:|:---------:|:------:|:------:|:------:|
> | Meteor          | 77.9 |   74.9  |    53.4   |  57.3  |  87.1  |  52.8  |
> | Meteor-LLaVA-HR | 80.8 |   77.9  |    57.4   |  59.5  |  90.2  |  54.0  |
>
> Aligned with the results in MM1 [R1], increasing image resolution significantly improves vision-language performance. From these two tables, it is evident that enhancing image resolution with Meteor-Mamba is more effective than merely enlarging the Meteor-Mamba architecture size.
>
> ---
>
> **Q3. To be honest, the writing of the paper should be improved. I take a lot of time to understand the technical details. The method should be better organized.**
>
> **A3.** We appreciate your feedback regarding the organization of the technical details. In the revised version of our manuscript, we will reorganize the technical details and the methods section to improve clarity and understandability.
>
> ---
>
> **References**
>
> [R1] McKinzie, Brandon, et al. "Mm1: Methods, analysis & insights from multimodal llm pre-training." arXiv preprint arXiv:2403.09611 (2024).
>
> ---
>
> **We hope this rebuttal can improve the "Rating" score more positively.**

---

> ### Comment · Reviewer_Hj2J · 2024-08-09
>
> Thanks for authors'rebuttal. The provided results further address my concerns. The newly added comparisons should be included in the final version.  Therefore, I would like to improve my score to accpet.

---

### Official Review · Reviewer_iRmr · 2024-07-11

**Soundness:** 3
**Presentation:** 3
**Contribution:** 3
**Rating:** 6
**Confidence:** 4

**Summary:**

The paper "Meteor: Mamba-based Traversal of Rationale for Large Language and Vision Models" presents a new efficient large language and vision model (LLVM), Meteor, which leverages multifaceted rationale to enhance understanding and answering capabilities. The paper  introduces a new concept of traversal of rationale, demonstrating the effectiveness of Meteor in embedding lengthy rationales containing abundant information and improving vision language performances across various evaluation benchmarks requiring diverse capabilities.

**Strengths:**

1. The paper introduces a novel concept of traversal of rationale and effectively demonstrates the efficiency of embedding lengthy rationales using the Mamba architecture.
2. The paper provides comprehensive performance comparison with existing LLVMs in Figure 1, 2 and Table 1. Based on this, the experimental results and ablation studies provide strong evidence of the effectiveness of Meteor in improving vision language performances across diverse evaluation benchmarks.
3. The paper provides a detailed description of the model architecture, training strategy, and evaluation benchmarks, ensuring reproducibility and understanding of the proposed method.

**Weaknesses:**

Writing:
1. All Figures (fonts, line widths, etc.) in the paper are not uniform, there is too much white space. And even Figure 1 shows strange tables and font proportions (fonts are also not clear).

Motivation:
2. The paper employ the Mamba architecture to embed lengthy rationales containing abundant information, but without setting and demonstrating task scenarios that existing LVLMs does not achieve that require extreme long sequence modeling capabilities.
3. It appears to share the same motivation as some existing methods, for instance, Vary-toy[1]. Please provide a more detailed explanation of how the proposed "traversal of rationale" differs from the design implemented in Vary-toy.

Method:
4. Show more compassion with the MMLM models based on Mamba, like Cobra[2] or VL-Mamba[3].

[1] Small Language Model Meets with Reinforced Vision Vocabulary
[2] Cobra: Extending Mamba to Multi-Modal Large Language Model for Efficient Inference
[3] VL-Mamba: Exploring State Space Models for Multimodal Learning

**Questions:**

Please refer to the weakness

---

> ### Author Rebuttal · Authors · 2024-07-31
>
> Thank you for your insightful comments! We will incorporate the following rebuttal into our manuscript to enhance overall understanding in the potential camera-ready version.
>
> ---
>
> **Q1. Writing: All Figures (fonts, line widths, etc.) in the paper are not uniform, there is too much white space. And even Figure 1 shows strange tables and font proportions (fonts are also not clear).**
>
> **A1.** We will ensure uniform font formats and line widths for all figures and remove unnecessary white space as you suggested. Additionally, we will correct the tables and font proportions in Figure 1 to improve clarity.
>
> ---
>
> **Q2. Motivation: The paper employ the Mamba architecture to embed lengthy rationales containing abundant information, but without setting and demonstrating task scenarios that existing LVLMs does not achieve that require extreme long sequence modeling capabilities.**
>
> **A2.** We would like to clarify that our focus is not on generating long rationale text, but on embedding multifaceted information used in larger models with a variety of forms. This includes fundamental image understanding, real-world knowledge about common sense and non-object concepts, and step-by-step procedures for solving complex questions.
>
> For detailed reasoning capabilities of Meteor, please refer to Answer 2 of "Reviewer XZUR," which describes Meteor's complex reasoning capabilities on challenging benchmarks such as MM-Vet and LLaVA-W.
>
> ---
>
> **Q3. It appears to share the same motivation as some existing methods, for instance, Vary-toy. Please provide a more detailed explanation of how the proposed "traversal of rationale" differs from the design implemented in Vary-toy.**
>
> **A3.** Vary-toy generates a new vision vocabulary network trained on a smaller language model, which is later merged with CLIP to train a larger language model. Consequently, Vary-toy aims to achieve more fine-grained vision perception through an expanded vision tokenizer.
>
> In contrast, our design of traversal of rationale is motivated by the goal of embedding multifaceted rationale for not only vision perception but also common sense, non-object concepts, and step-by-step procedures. Specifically, we employ the Mamba architecture to embed long sequential rationales into $<tor>$ tokens, which are subsequently propagated into Meteor-MLM. These tokens plays a role in conveying embeded multifaceted rationale into Meteor-MLM which ultimately answers the question.
>
> ---
>
> **Q4. Method: Show more comparison with the MMLM models based on Mamba, like Cobra or VL-Mamba.**
>
> **A4.** The table below compares our model to Cobra, VL-Mamba, RoboMamba [R1], ML-Mamba [R2]. We will include the results of these Mamba-based LLVMs in the updated manuscript.
>
> | LLVMs     |   VQAv2  |    GQA   |  SQA-IMG |  TextVQA |   POPE   |    MMB   |  MM-Vet  |
> |-----------|:--------:|:--------:|:--------:|:--------:|:--------:|:--------:|:--------:|
> | Cobra     |   76.9   |   59.9   |     -    |   57.9   |   88.2   |     -    |     -    |
> | VL-Mamba  |   76.6   |   56.2   |   65.4   |   48.9   |   84.4   |   57.0   |   32.6   |
> | RoboMamba |   79.1   |   64.4   |     -    |     -    |   86.9   |   65.7   |   29.7   |
> | ML-Mamba |   75.3    |   60.7   |     -    |     52.2    |   88.3   |   -   |   -   |
> | Meteor    | **82.5** | **64.7** | **88.3** | **67.5** | **88.7** | **82.9** | **57.3** |
>
> ---
>
> **References**
>
> [R1] Liu, Jiaming, et al. "RoboMamba: Multimodal State Space Model for Efficient Robot Reasoning and Manipulation." arXiv preprint arXiv:2406.04339 (2024).
>
> [R2] Huang, Wenjun, and Jianguo Hu. "ML-Mamba: Efficient Multi-Modal Large Language Model Utilizing Mamba-2." arXiv preprint arXiv:2407.19832 (2024).
>
> ---
>
> **We hope this rebuttal can improve the "Rating" score more positively.**

---

> > ### Comment · Reviewer_iRmr · 2024-08-13
> > **Official Comments by Reviewer iRmr**
> >
> > The rebuttal has addressed part of my questions. About Q3, many existing methods apply the pre-encoding method to achieve united representation.  Vary-toy is just one example of this idea. So i still think the contribution is a little bit limited. Considering the impressive results, i will still maintain my score.

---

> > > ### Author Response · Authors · 2024-08-14
> > > **Response**
> > >
> > > We agree with the pre-encoded system is a common structure, but we disagree with that our work seems limited contribution. We newly presented the pre-embedded concept of "multifaceted rationale". The key innovation lies not in the structure itself—whose significance is widely recognized—but in the nature of what is embedded. We think embedding new reasonable thing is really novel where we introduce new technique of "traversal of rationale".

---

### Official Review · Reviewer_XZUR · 2024-07-22

**Soundness:** 3
**Presentation:** 3
**Contribution:** 3
**Rating:** 5
**Confidence:** 4

**Summary:**

This paper introduces a novel approach to enhance the performance of large language and vision models (LLVMs). The proposed model, Meteor, leverages multifaceted rationale through the Mamba architecture to efficiently embed lengthy (latent) rationales and improve understanding and answering capabilities across diverse vision language benchmarks. The key innovation lies in the traversal of rationale concept and the efficient processing capabilities of the Mamba architecture, which allows for handling sequential data with linear time complexity.

**Strengths:**

Well-Written: The paper is clearly written, with a logical flow of ideas and a well-structured presentation of the methodology and results.

Innovative Approach: The concept of traversal of rationale and the use of the Mamba architecture for rationale embedding are novel and well-motivated.

Effective Performance: The proposed approach achieves significant improvements in vision language performances across multiple benchmarks, demonstrating its effectiveness.

**Weaknesses:**

i) Figure 3 (Stage-One Pre-Training):

More clarity and intuition behind the model design are needed. Given that the training is formulated as an autoregressive token generation task:
a) Are TOR embeddings predicted as special tokens [TOR]?
b) Or are losses on these tokens bypassed?
If the latter, a potential drawback is the inability to assess the model's performance after the first stage of training, as it cannot produce complete sentences/rationales.


ii) Benchmarks and Reasoning Capabilities:

The proposed benchmarks may not sufficiently test complex rationale reasoning (e.g., current tasks are mostly recognition or OCR-heavy).
Recommendation: Conduct further studies in the final version on reasoning-oriented benchmarks such as Visual Commonsense Reasoning (VCR) [r1] to substantiate claims of:
a) Enhanced reasoning capability,
b) More grounded results (less hallucination) compared to existing literature.
Note: While VCR includes rationale annotations, evaluating only the answer component would be sufficient, given the proposed method's current limitations in directly sampling rationale output.


iii) Ablation Studies (Table 3a):

Clarification needed: Did the baselines undergo the same first-stage pre-training as Mamba? If so, what were the training objectives for each method?


iv) Comparison Methodology (Figure 2):

Line 147 indicates that the training data includes annotations from ChartQA and AI2D. Some results (e.g., Gemini-Pro and GPT-4V) are zero-shot and not directly comparable.
Recommendation: Revise the plots to accurately reflect these differences in training/evaluation conditions.


Reference:
[1] Zellers et al., From Recognition to Cognition: Visual Commonsense Reasoning, CVPR 2019.

**Questions:**

see previous sections

**Limitations:**

appear to be sufficient

---

> ### Author Rebuttal · Authors · 2024-07-31
>
> Thank you for your valuable comments. We will incorporate the following rebuttal into our manuscript to enhance overall understanding in the potential camera-ready version.
>
> ---
>
> **Q1. Figure 3 (Stage-One Pre-Training): More clarity and intuition behind the model design are needed. Given that the training is formulated as an autoregressive token generation task: a) Are TOR embeddings predicted as special tokens [TOR]? b) Or are losses on these tokens bypassed? If the latter, a potential drawback is the inability to assess the model's performance after the first stage of training, as it cannot produce complete sentences/rationales.**
>
> **A1.** Among options a) and b), the correct approach is b). This means that special tokens $<tor>$ are not predicted tokens in the autoregressive loss but are instead input tokens in Meteor-Mamba. These tokens play a crucial role in embedding the multifaceted rationale interleaved between the special tokens $<tor>$. In other words, the model learns to predict the partial segments of a multifaceted rationale that are between the $<tor>$ tokens in the autoregressive loss.
>
> As you pointed out, there may be a potential drawback in "Meteor-Mamba" due to its inability to produce full rationale sentences. However, we argue that "Meteor-MLM" is ultimately used to generate text responses, so the text generation quality of "Meteor-MLM" reflects on the reasoning capability learned from "Meteor-Mamba." We have provided detailed validation results in Answer 2 of Question 2, as below.
>
> ---
>
> **Q2. Benchmarks and Reasoning Capabilities: The proposed benchmarks may not sufficiently test complex rationale reasoning (e.g., current tasks are mostly recognition or OCR-heavy). Recommendation: Conduct further studies in the final version on reasoning-oriented benchmarks such as Visual Commonsense Reasoning (VCR)**
>
> **A2.** We respectfully disagree with the assertion that "the proposed benchmarks may not sufficiently test complex rationale reasoning." The evaluation benchmarks MM-Vet and LLaVA-W, as presented in Tables 1-3, already include challenging measures of complex reasoning based on GPT-4 evaluation.
>
> For MM-Vet, the sub-benchmark "Language Generation" in Table 2(d) specifically represents the quality of text generation required to answer complex questions. Additionally, LLaVA-W contains a sub-benchmark explicitly titled "Complex Reasoning." Below, we present a table showing the sub-benchmarks of MM-Vet and LLaVA-W compared with Meteor and the previous SOTA LLVMs within equal size (Meteor uses about 2 times more dataset samples than them. In addition, they are not performant well to these challenging benchmarks.). This table also includes an ablation study where "Meteor-Mamba" is removed, highlighting the effectiveness of "Meteor-Mamba" in enhancing reasoning capabilities.
>
> | LLVMs                    | Recognition |  OCR | Knowledge | Language Generation | Spatial Awareness | Math Problems | Avg  |
> |--------------------------|:-----------:|:----:|:---------:|:-------------------:|:-------------------:|:---------------:|:------:|
> | CoLLaVO                  |     45.6    | 31.1 |    29.8   |         30.2        | 37.9              | 5.8           | 41.0 |
> | MoAI                     |     48.3    | 34.8 |    33.5   |         33.0        | 39.7              | 7.7           | 43.7 |
> | Meteor w.o. Meteor-Mamba |     44.5    | 33.5 |    41.8   |         31.3        | 38.6              | 29.2          | 44.8 |
> | Meteor                   |     54.1    | 60.1 |    44.2   |         45.0        | 59.3              | 57.7          | 57.3 |
>
> | LLVMs                    | Conversation | Detail description | Complex Reasoning |  Avg |
> |--------------------------|:------------:|:------------------:|:-----------------:|:----:|
> | CoLLaVO                  |     51.1     |        73.8        |        77.1       | 69.5 |
> | MoAI                     |     48.5     |        76.0        |        80.6       | 71.9 |
> | Meteor w.o. Meteor-Mamba |     67.4     |        72.9        |        75.2       | 73.7 |
> | Meteor                   |     80.3     |        87.2        |        91.2       | 87.1 |
>
> We believe the reasoning questions in MM-Vet and LLaVA-W are significantly more complex than those in the Visual Commonsense Reasoning (VCR) benchmark you recommended. Recent models, such as OV-Grounding and GPT4RoI, have nearly matched human performance on the VCR leaderboard, indicating the need for more challenging questions, such as those found in MM-Vet and LLaVA-W. These benchmarks remain unsaturated and continue to pose difficult problems evaluated using GPT-4, ensuring a rigorous assessment of reasoning capabilities.
>
> ---
>
> **Q3. Ablation Studies (Table 3a): Clarification needed: Did the baselines undergo the same first-stage pre-training as Mamba? If so, what were the training objectives for each method?**
>
> **A3.** Yes, all baselines underwent the same first-stage pre-training as Mamba. The training objectives for each method were identical: "The partial segments of a multifaceted rationale between special tokens $<tor>$ are learned to predict in autoregressive loss, where the tokens $<tor>$ are not predicted tokens in autoregressive loss."
>
> ---
>
> **Q4. Comparison Methodology (Figure 2): Line 147 indicates that the training data includes annotations from ChartQA and AI2D. Some results (e.g., Gemini-Pro and GPT-4V) are zero-shot and not directly comparable. Recommendation: Revise the plots to accurately reflect these differences in training/evaluation conditions.**
>
> **A4.** For Gemini-Pro and GPT-4V, the training datasets are unknown as they are totally closed-source LLVMs, making it uncertain whether their results are truly zero-shot. However, for Table 1, we will explicitly mention that the ChartQA and AI2D performances for LLaVA family models are evaluated under zero-shot conditions. This clarification will help accurately reflect the differences in training and evaluation conditions.
>
> ---

---

> > ### Comment · Reviewer_XZUR · 2024-08-12
> > **Questions have been addressed in the rebuttal**
> >
> > The rebuttal has addressed the questions from the initial review and the authors indicated corresponding revisions in the final version. Therefore, I maintain my original positive review.

---

### Author Response · Authors · 2024-08-10
**Wait for Response**

The three reviewers: XZUR, iRmr, and Hj2J scored our paper to positively "5", "5", "7", while the one reviewer zsWf scored negatively "4". We hope our rebuttals make the reviewers who do not respond yet possibly improve their scores. If the discussion should be needed more, then we will be sincerely and rapidly participating in the future discussion. Kindly note that, we would like to respectfully thank the reviewer Hj2J for rapid response.

---

### Decision · Program_Chairs · 2024-09-25

**Decision:**

Accept (poster)

**Comment:**

The paper received Borderline Accept, Weak Accept, Accept and Borderline Accept.

The main contribution of this paper is a new pretrained vision-and-language model that demonstrates good results across a range of benchmarks and outperforms a recent LLaVA-Next competitor. Reviewers were mostly positive and improved scores after rebuttal discussions. One concern that was voiced in a couple of reviews was the clarity on how the method compares in terms of inference overhead and other aspects besides accuracy on downstream benchmarks. Authors are encouraged to incorporate these additional comparisons in their final version.